# Advancing quantum imaging through learning theory

Yunkai Wang [1,2,3] ✉, Changhun Oh [4], Junyu Liu [5,6], Liang Jiang [6] ✉ &
Sisi Zhou [1,2,3,7] ✉

We study quantum imaging by applying the resolvable expressive capacity (REC) formalism developed for physical neural networks (PNNs). In this paradigm of quantum learning, the imaging system functions as a physical learning device that maps input parameters to measurable features, while complex practical tasks are handled by training only the output weights, enabled by the systematic identification of well-estimated features (eigentasks) and their corresponding sample thresholds. Using this framework, we analyze both direct imaging and superresolution strategies for compact sources, defined as sources with sizes bounded below the Rayleigh limit. In particular, we introduce the orthogonalized SPADE method—a nontrivial generalization of existing superresolution techniques—that achieves superior performance when multiple compact sources are closely spaced. This method relaxes the earlier superresolution studies' strong assumption that the entire source must lie within the Rayleigh limit, marking an important step toward developing more general and practically applicable approaches. Using the example of face recognition, which involve complex structured sources, we demonstrate the superior performance of our orthogonalized SPADE method and highlight key advantages of the quantum learning approach—its ability to tackle complex imaging tasks and enhance performance by selectively extracting well-estimated features.

The quality of an image formed by a single-lens system depends on several factors, including the lens's resolution, the measurement strategy employed in the image plane, and the number of collected samples. Notably, it has been shown that by optimizing the measurement design, one can resolve two-point sources within the Rayleigh limit, surpassing the Rayleigh's criterion[1]. The concept of superresolution has been extended in various directions[2], focusing on imaging a single compact source—an object much smaller than the Rayleigh limit. These extensions include more careful treatment of the measurement and data analysis for two point sources[3,4], general

sources within the Rayleigh limit[5–7], sources beyond the weak-source limit[8–10], and point sources in higher dimensions[11–13]. These theories have been experimentally demonstrated for estimating point sources under various scenarios[14–21] and for estimating source moments[20].

Superresolution often relies on conventional statistical tools, such as the Fisher-information-matrix (FIM) approach for quantifying parameter-estimation precision[2–21], and on the Chernoff bound or likelihood-ratio method for discrimination tasks[22–26]. However, these conventional statistical tools face at least two fundamental challenges when applied to complex imaging tasks. (i) Modeling complexity and

[1]Perimeter Institute for Theoretical Physics, Waterloo, ON, Canada. [2]Department of Applied Mathematics, University of Waterloo, Waterloo, ON, Canada. [3]Institute for Quantum Computing, University of Waterloo, Waterloo, ON, Canada. [4]Department of Physics, Korea Advanced Institute of Science and Technology, Daejeon, Republic of Korea. [5]Department of Computer Science, The University of Pittsburgh, Pittsburgh, PA, USA. [6]Pritzker School of Molecular Engineering, The University of Chicago, Chicago, IL, USA. [7]Department of Physics and Astronomy, University of Waterloo, Waterloo, ON, Canada. ✉e-mail: ywang10@perimeterinstitute.ca; liangjiang@uchicago.edu; sisi.zhou26@gmail.com

ambiguity. It is often unclear which parameterization is most appropriate for a practical imaging task. Image moments, Fourier coefficients, and pixel intensities all provide complete representations but lead to different interpretations and performance. In addition, the likelihood-ratio method requires a full statistical model of the object, which is extremely difficult to construct for complex tasks—for example, in face recognition. (ii) Finite-sample reliability. With limited data, only a subset of features can be estimated accurately; many others are dominated by noise. Effective use of the data therefore requires identifying and retaining only well-estimated features for downstream analysis—an especially difficult step in imaging, where the number of parameters is, in principle, infinite.

The machine learning approach has achieved tremendous success in imaging applications, enabling the handling of complex practical tasks. In this work, we implement imaging tasks with a paradigm of quantum learning—physical neural networks (PNNs)[27–40]—following in particular the resolvable expressive capacity (REC) formalism in Ref. [40] to address the limitations of the FIM approach. The quantum learning approach adopted here encompasses both model training and inference using the trained model. PNNs encode inputs—such as the positions of incoherent point sources—into an analog physical system whose evolution is fixed and governed by its underlying physics, mapping the inputs to the measured high-dimensional features. A key advantage is that practical tasks can be performed by training only the output weight—equivalent to applying linear or logistic regression to the measured features—while leaving the internal structure unchanged. The relationship between the specific system dynamics and the class of tasks it can realize is a fundamental question in PNNs. The REC formalism addresses this question by identifying the resolvable features and quantifying their achievable precision in the presence of finite-sample noise[40]. The imaging system is an analog physical system that naturally functions as a learning device in PNNs, with output weights trained to perform imaging tasks. And applying the REC formalism to imaging identifies well-estimated features through eigentasks that are invariant under parameterization, thereby guiding the formulation of complex practical tasks for a given imaging system and prior information, directly addressing the first challenge faced by conventional statistical tools. Moreover, it provides a way to estimate the sample threshold required to detect each eigentask. This makes it particularly useful for determining which measured features can be reliably used in downstream analysis by selecting low-noise eigentasks based on a threshold. This key advantage, highlighted in the original paper[40], can enhance the performance of diverse machine learning tasks, such as classification, regression, and clustering, and addresses the second challenge faced by conventional statistical tools. A more detailed introduction to the PNNs is provided in Supplementary Note 1 B.

Besides adopting a quantum learning perspective to study the imaging problem, we extend the existing discussions on superresolution[1–22] to the broader challenge of imaging multiple compact sources, each individually constrained within the Rayleigh limit, or equivalently general sources that exceed the Rayleigh limit while exhibiting clustered substructures. This motivation arises as follows: prior studies typically assume that the whole source lies within the Rayleigh limit, i.e., it is a single compact source. However, in practical imaging, sources are often larger than the Rayleigh limit, while we aim to resolve fine features within the source that are below the Rayleigh limit. To address this, a natural idea is to partition the source into compact regions and apply superresolution techniques locally, effectively treating the problem as imaging multiple compact sources. But a straightforward application of superresolution to individual compact sources, referred to as the separate SPADE method, fails to offer an advantage over direct imaging. This is because nearby sources, when separated by distances not much greater than the width of the point spread function (PSF), introduce additional noise into the

measurement. However, our generalized approach to superresolution, called the orthogonalized SPADE method, can achieve superresolution even when the separation between compact sources is as small as the PSF width. This discussion advances superresolution to imaging multiple reasonably nearby compact sources, constituting a nontrivial generalization of the earlier SPADE method and a step toward making the approach more practical.

## Results
### Preliminary
We now provide a more detailed explanation of how imaging tasks are realized using PNNs. The imaging system can be regarded as an input-output map, where the input is a set of system parameters $\boldsymbol{\theta}$ (e.g., the locations of point sources) and the output is a set of measured features, which are the probabilities $P_j(\boldsymbol{\theta}) = Tr[\rho(\boldsymbol{\theta})M_j]$, where $M_j$ denotes the $j$th element of the positive operator-valued measure (POVM). A general learning task—such as classification or regression—can be formulated in terms of a target function $f(\boldsymbol{\theta})$ of the input parameters. Realizing an imaging task is then equivalent to approximating this target function using the measured features, which corresponds to training the output weights of the PNNs[27–39].

For a given physical system, it is important to determine the class of functions that can be approximated using the measured features. Moreover, due to sampling noise, the quantities $\overline{P}_j(\boldsymbol{\theta})$ can only be estimated approximately. It is also necessary to quantify the precision with which these functions can be approximated. These questions, concerning the capability of a physical system when regarded as a PNN, are addressed by the recently developed REC formalism[36,40]. REC formalism analyzes the resolvable function space of a physical system under sample noise via the concept of REC,

$$C[f] = 1 - \min_{W} \frac{\mathbb{E}_{\boldsymbol{\theta}}\left[\mathbb{E}_{\mathcal{X}}\left[\left(\sum_j W_j \overline{P}_j(\boldsymbol{\theta}) - f(\boldsymbol{\theta})\right)^2\right]\right]}{\mathbb{E}_{\boldsymbol{\theta}}[f(\boldsymbol{\theta})^2]}, \tag{1}$$

where we take the expectation value for the output samples $\mathcal{X}$ and the prior distribution $p(\boldsymbol{\theta})$, $f(\boldsymbol{\theta})$ is approximated by a linear combination of measured functions $\sum_j W_j \overline{P}_j(\boldsymbol{\theta})$, $W_j$ represents the weight coefficient in front of $\overline{P}_j(\boldsymbol{\theta})$ to be optimized to achieve the optimal linear approximation of $f(\boldsymbol{\theta})$, where the index $j$ corresponds to different measurement outcomes. REC $C[f]$, which takes values between 0 and 1, can be understood as the normalized mean-squared accuracy of approximating $f(\boldsymbol{\theta})$, where $C[f] = 1$ represents a perfect approximation using the measured features, and deviation from 1 indicates that the target function cannot be well approximated.

To quantify the overall performance of an imaging system as PNNs, we are interested in identifying the set of functions $f(\boldsymbol{\theta})$ that can be approximated in this way and in quantifying the effective size of this set, given the physical imaging system, the finite number of samples $S$, and prior knowledge about the input $\boldsymbol{\theta}$. The total REC $C_T := \sum_k C[g_k]$[40] fulfills this purpose, where $\{g_k\}_k$ can be any complete orthonormal basis of functions in the Hilbert space equipped with inner product $\mathbb{E}_{\boldsymbol{\theta}}[g_k(\boldsymbol{\theta})g_\ell(\boldsymbol{\theta})]$. The value of total REC can be obtained from the following eigenvalue problem,

$$D_{kj} = \delta_{kj}Tr\{M_k\widehat{\rho}^{(1)}\}, \quad G_{jk} = Tr\{(M_j \otimes M_k)\widehat{\rho}^{(2)}\}, \tag{2}$$

$$\widehat{\rho}^{(t)} = \mathbb{E}_{\boldsymbol{\theta}}[\rho(\boldsymbol{\theta})^{\otimes t}], \tag{3}$$

$$V = D - G, \quad Vr_k = \beta_k^2 Gr_k, \tag{4}$$

where $\beta_k^2$ and $r_k$ are the $k$th eigenvalue and eigenvector (in increasing order). The eigenbasis $r_k$ correspond to a minimal set of eigentasks

$f_k(\boldsymbol{\theta}) := \sum_m r_{km} P_m(\boldsymbol{\theta})$ that saturate the available REC of the system in the space of all functions of input parameters $\boldsymbol{\theta}$. Then

$$C_T = \sum_k C[f_k] = \sum_k \frac{1}{1 + \beta_k^2/S}, \qquad (5)$$

where $S$ is the number of samples. Intuitively, $C_T$ quantifies how many independent features of the underlying signal can be captured by the measurement process, and capturing more features improves performance on complex learning tasks by providing greater freedom to approximate the target function $f(\boldsymbol{\theta})$. Treating the imaging system as a PNN involves first identifying the eigentasks in the REC formalism and then training the output weights—essentially performing logistic or linear regression using the values of eigentasks $f_k(\boldsymbol{\theta})$—while including only the well-estimated eigentasks under a finite sample size and discarding noisy ones to ensure optimal performance. Another important property of this formalism is that reparameterization leaves both the total REC and the eigentasks unchanged (see Methods section), making the strategy independent of any artificial parameterization and determined solely by the physical system and the structure of the problem. A more detailed introduction to this formalism, proposed in ref. 40, is provided in Supplementary Note 1 C.

When we apply this quantum learning approach to the super-resolution setting of imaging multiple compact sources, as shown in Fig. 1, we observe that the images exhibit two types of features: Rayleigh resolvable features and sub-Rayleigh features. The Rayleigh resolvable features are determined by the intensity of each compact source, which can be measured with a constant number of samples independent of the source size for both direct imaging and super-resolution methods. In contrast, the sub-Rayleigh features involve details below the Rayleigh limit, requiring the number of samples to scale inversely with the source size, where carefully designed super-resolution methods demonstrate clear advantages. Intuitively, the total REC quantifies the number of reliably estimated features—both large and small—and increases as smaller features become resolvable. We first examine the distinctive behavior of the eigenvalues $\beta_k^2$ and the total REC $C_T$ in each case of superresolution, and introduce our orthogonalized SPADE method. We then demonstrate the advantages of both this quantum learning approach and our new orthogonalized SPADE method through a concrete learning task.

### Resolving two-point sources

As a simplest example, we begin with the imaging of two incoherent point sources in one dimension. A single photon received on the image plane can be described as $\rho(L) = \frac{1}{2}(|\psi_1\rangle\langle\psi_1| + |\psi_2\rangle\langle\psi_2|)$, where $|\psi_i\rangle = \int du\psi(u - u_i)|u\rangle$, $|u\rangle = a_u^\dagger|0\rangle$ is the single photon state at position $u$, and we choose the PSF $\psi(u) = \exp(-u^2/4\sigma^2)/(2\pi\sigma^2)^{1/4}$. Define the separation $L$, which is the input of learning task $\theta = L$ and assume $u_2 = L/2$, $u_1 = -L/2$. To enable analytical analysis, we focus on the binary SPADE measurement, which is capable of achieving super-resolution in resolving two-point sources as introduced in ref. 1, where POVM $M_0 = |\phi_0\rangle\langle\phi_0|$, $M_1 = I - M_0$, $|\phi_0\rangle = \int du\phi_0(u)|u\rangle$, $\phi_0(u) = \frac{1}{(2\pi\sigma^2)^{1/4}}\exp(-\frac{u^2}{4\sigma^2})$. Assuming the prior knowledge about the separation is described as $p(L) = \frac{1}{\sqrt{2\pi}\gamma}\exp(-\frac{L^2}{2\gamma^2})$, we can calculate the total REC, $C_T$, which here represents the total number of linearly independent functions $f(L)$ that can be expressed as a linear combination of the measured probabilities $P_k = tr(\rho(L)M_k)$. Assuming $\gamma \ll \sigma$ to exhibit the advantage of superresolution within the Rayleigh limit, we find that

$$\begin{aligned}\beta_0^2 &= 0, \\ \beta_1^2 &= \frac{8}{\alpha^2} + \frac{3}{4} - \frac{1}{64}\alpha^2 + O(\alpha^4), \quad \alpha = \gamma/\sigma,\end{aligned} \qquad (6)$$

where $\alpha$ is roughly the ratio between the separation and width of the PSF and $\alpha \ll 1$ when the two-point sources are very close to each others.

We can compare this with the direct imaging case, where we directly project onto each spatial mode $E_x = |x\rangle\langle x|_x$. In this case, we find that $\beta_0^2 = O(1)$, $\beta_1^2 = \Theta(\alpha^{-4})$, and $\beta_2^2 = \Theta(\alpha^{-8})$ after discretizing the spatial coordinate. More details of the calculations for both the direct imaging and SPADE methods are provided in Supplementary Note 2. The much larger eigenvalue $\beta_1^2$ in direct imaging indicates poorer performance compared to binary SPADE, as it requires a larger number of samples $S$ to achieve the same $C_T$.

### Resolving a single compact source

We now consider the problem of imaging a single compact source, defined as a generally distributed source whose spatial extent is bounded well below the Rayleigh limit. This represents the most general setting for applying superresolution in imaging that has been considered in previous works[2–22]. Assume the normalized source intensity $I(u)$ is confined within the interval $[-L/2, L/2]$. We can define the moments as $\int du\, I(u)(\frac{u-u_0}{L})^n = x_n$, which completely describe the source and are the input for the learning task $\boldsymbol{\theta} = \vec{x} = [x_0, x_1, x_2, \cdots]$. Within the Rayleigh limit, $\alpha = L/\sigma \ll 1$, where $\sigma$ represents the width of the PSF, the size of the compact source is significantly smaller than the resolution limit. For any prior $p(\vec{x})$ and PSF, we find that for the direct

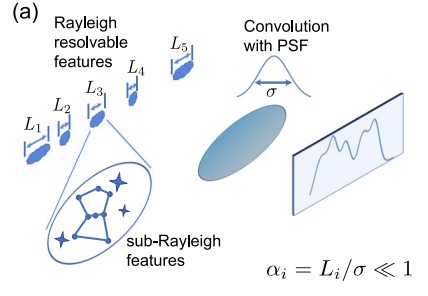
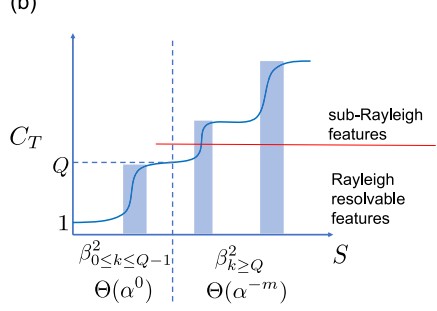

**Fig. 1 | Imaging of both Rayleigh resolvable features and sub-Rayleigh features. a** Multiple compact sources of size $L_i$ are imaged by a lens with a PSF width of $\sigma$. Besides Rayleigh-resolvable features, we would also like to extract information from sub-Rayleigh features, which are associated with the small parameter $\alpha = L/\sigma$. **b** The total REC, $C_T$, which shows a stepwise increase, is plotted as a function of the number of samples, $S$. The threshold of $S$ for each stepwise increase of $C_T$ in the shaded region is determined by the eigenvalue $\beta_k^2$ associated with the corresponding eigentask in learning. Each time $C_T$ increases by 1, there is a

corresponding eigenvalue $\beta_k^2$, with the sample number threshold following $S \sim \Theta(\beta_k^2)$. The intensity of each compact source corresponds to resolvable features, which can be imaged with a constant number of samples, scaling as $S \sim \beta_{0 \le k \le Q-1}^2 = \Theta(\alpha^0)$, independent of the source size. In contrast, sub-Rayleigh features that reveal detailed information about each compact source require a number of samples scaling inversely with the source size, following $S \sim \beta_{k \ge Q}^2 = \Theta(\alpha^{-m})$, where $\alpha$ is determined by the compact sources, and $m$ depends on the order of moments.

imaging

$$\beta_0^2 = 0, \; \beta_1^2 = \Theta(\alpha^{-2}), \; \beta_2^2 = \Theta(\alpha^{-4}), \; \cdots \quad (7)$$

where $\beta_0^2 = 0$ is a trivial eigenvalue which corresponds to the fact that $\sum_m P_m = 1$.

For superresolution, we adopt the measurement construction from ref. 7, as reviewed in Supplementary Note 1 A. For simplicity, we still refer to this method as the SPADE method throughout the discussion. Intuitively, the SPADE method isolates higher-order moments by constructing probability distributions that start at higher-order terms of $\alpha$, which serves as the signal strength, thereby suppressing lower-order terms of $\alpha$ that act as noise. This yields a much better signal-to-noise ratio for estimating those moments, especially in the weak-signal regime. For any prior $p(\vec{x})$ and PSF, we find that for the SPADE method

$$\beta_0^2 = 0, \; \beta_1^2 = \Theta(\alpha^{-2}), \; \beta_2^2 = \Theta(\alpha^{-2}),$$
$$\beta_3^2 = \Theta(\alpha^{-4}), \; \beta_4^2 = \Theta(\alpha^{-4}), \; \cdots \quad (8)$$

Compared to direct imaging, the SPADE method achieves smaller $\beta_k^2$, which significantly reduces the required $S$ to achieve the same $C_T$.

We demonstrate the significance of $\beta_k^2$ as the threshold for the stepwise increase in the total REC $C_T$, as shown in Fig. 2. The total REC $C_T = \sum_k \frac{1}{1 + \beta_k^2/S}$ shows that each $\beta_k^2$ sets the sample size at which its eigentask contributes significantly, with contributions near 1 when $S \gg \beta_k^2$ and negligible when $S \ll \beta_k^2$. For direct imaging, $C_T$ increases by 1 at each step, while for the SPADE method, $C_T$ increases by 2 per step, as expected. As $\alpha$ decreases, the plateau regions expand. All numerical calculations in this work assume a Gaussian PSF $\psi(u) = \exp(-u^2/4\sigma^2)/(2\pi\sigma^2)^{1/4}$; however, our method is applicable to any PSF. We choose the prior distribution for the moment vectors $\vec{x}$ by randomly generating a set of images and assuming they occur with equal probability, thereby establishing $p(\vec{x})$ as the empirical distribution of the resulting moment vectors, as detailed in Supplementary Note 6. For illustration purpose, we plot the data for only one instance of the randomly generated prior distributions (and in all figures below where the prior is picked randomly).

Note that the threshold of $S$ is not precisely located at $\alpha^{-2n}$ in each case. This deviation arises from a constant prefactor in $\beta_k^2$. This constant prefactor is independent of $\alpha$ and is $\sim 10^2$ in Fig. 2. This is reasonable because, even in the simpler imaging task where sources are extended outside the Rayleigh limit (i.e., $\alpha \gg 1$), hundreds or more samples are still required to effectively image a source. The prefactors depend on the imaging strategy and the prior information. We provide a more detailed discussion of these prefactors in Supplementary Note 7.

The $n$th eigentask corresponds to $\sum_m r_{nm} P_m$ as a function of $\vec{x}$, where $r_{nm}$ are the components of the eigenvectors obtained by solving Eq. (4), and the corresponding REC is given by $1/(1 + \beta_n^2/S)$. For direct imaging of a single compact source, we observe that the eigentasks converge to $x_n$ in the limit of small $\alpha$ to the leading order. For the SPADE method, we show that $r_{nm}$ becomes an triangular matrix as $\alpha \to 0$. The first two leading-order terms of both the $2k$-th and $(2k+1)$-th eigentasks have coefficients $x_{2k}$ and $x_{2k+1}$.

Further details on the derivation of the scaling of $\beta_k^2$ and the eigenvectors $r_n$, based on perturbation theory and confirmed by numerical calculations, are provided in Supplementary Note 3. So far, our discussion has focused on sources where the entire source lies within the Rayleigh limit. In this case, the total intensity, trivially equal to 1, is the Rayleigh resolvable feature that contributes to the total REC when only a constant number of samples (smaller than $1/\alpha^2$) is available. The sub-Rayleigh features contribute to the total REC when $\Omega(\alpha^{-2})$

samples are available. As we will see later, the Rayleigh resolvable features can become nontrivial when dealing with multiple compact sources.

## New superresolution methods on multiple compact sources

We now want to consider the scenario where we have multiple compact sources, each with a size within the Rayleigh limit, but collectively distributed over a region larger than the Rayleigh limit. The quantum state from these multiple compact sources is given by

$$\rho = \sum_{q=1}^{Q} \int du\, du_1\, du_2\, I_q(u)\psi(u - u_1)|u_1\rangle\langle u_2|\psi^*(u - u_2), \quad (9)$$

where $Q$ is the number of compact sources, $I_q(u)$ is the intensity distribution for $q$th compact source. We can expand near the centroid $u_q$ of $q$th source and reorganize the state as

$$\rho = \sum_{q=1}^{Q} \sum_{m,n=0}^{\infty} x_{m+n,q} \left| \psi_q^{(m)} \right\rangle \left\langle \psi_q^{(n)} \right|,$$
$$\left| \psi_q^{(m)} \right\rangle = \int du\, \psi_q^{(m)}(u) \left| u \right\rangle,$$
$$\psi_q^{(n)}(v) = \frac{\partial^n \psi(v - u)}{\partial u^n} \Big|_{u = u_q} \frac{L_q^n}{n!}, \quad (10)$$

where $L_q$ is the size (diameter) of $q$th source, $x_{n,q} = \int du\, I_q(u)\left(\frac{u - u_q}{L_q}\right)^n$ is the $n$th moment for the $q$th source and are the input for the learning task $\boldsymbol{\theta} = [x_{0,1}, x_{1,1}, x_{2,1}, \cdots, x_{0,Q}, x_{1,Q}, x_{2,Q}, \cdots]$. We find that for the direct imaging

$$\beta_0^2 = 0, \; \beta_{1 \le i \le Q-1}^2 = \Theta(1),$$
$$\beta_{Q \le i \le 2Q-1}^2 = \Theta(\alpha^{-2}), \; \beta_{2Q \le i \le 3Q-1}^2 = \Theta(\alpha^{-4}), \; \cdots \quad (11)$$

where $\alpha_q = \max L_q/\sigma$. Here, we assume that $L_q$ does not differ significantly, allowing the different $L_q$ values to be incorporated into the constant coefficients. It is then clear that there are $Q$ Rayleigh resolvable features corresponding to the intensity of each compact sources, and the number of sub-Rayleigh features also increases by a factor of $Q$. The scaling is numerically confirmed in Fig. 3(a) for the first six eigenvalues, and we expect $\beta_k^2 = \Theta(\alpha^{-2\lfloor k/Q \rfloor})$ to hold for eigenvalues with higher indices with any prior $p(\vec{x})$. In the numerical calculation, we assume two compact sources with centroids at $-L/4$ and $L/4$, with a random prior distribution for the moments (by randomly generating a set of images). For both $L = 2$ and $L = 20$, we observe the same scaling behavior in the direct imaging method.

To improve imaging performance, one could apply the SPADE method to each compact source individually—a technique referred to here as the separate SPADE method. Unfortunately, it only achieves the same scaling as direct imaging when the sources are not sufficiently spaced apart. This is because the proximity of other compact sources introduces significant noise when estimating higher-order moments. Alternatively, we can construct the orthonormal basis $|b_j^{(l)}\rangle$ using the Gram-Schmidt procedure, such that

$$\left\langle \psi_k^{(m)} \middle| b_j^{(l)} \right\rangle \begin{cases} = 0 & m \le l - 1 \\ = 0 & m = l \; \& \; k \le j - 1 \\ \neq 0 & otherwise \end{cases} \quad (12)$$

Choose POVM as the projection onto

$$\left| \phi_{j\pm}^{(l)} \right\rangle = \frac{1}{\sqrt{2}} \left( \left| b_j^{(l)} \right\rangle \pm \left| b_j^{(l+1)} \right\rangle \right), \quad (13)$$

where $j = 1, 2, 3, \cdots, Q$, $l = 0, 1, 2, \cdots, \infty$. The key intuition behind this construction is to ensure that when estimating the $x_{n,q}$ term in the

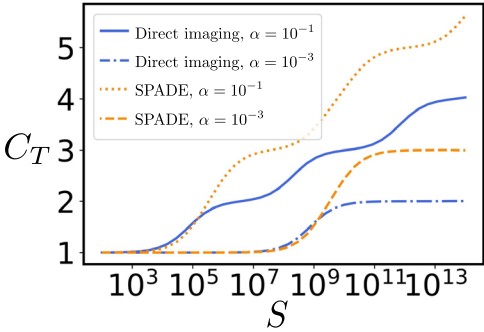

**Fig. 2 | Behavior of total REC for one compact source.** Total REC $C_T$ for direct imaging and the SPADE method as a function of $S$, when imaging one generally distributed compact source with different $\alpha$.

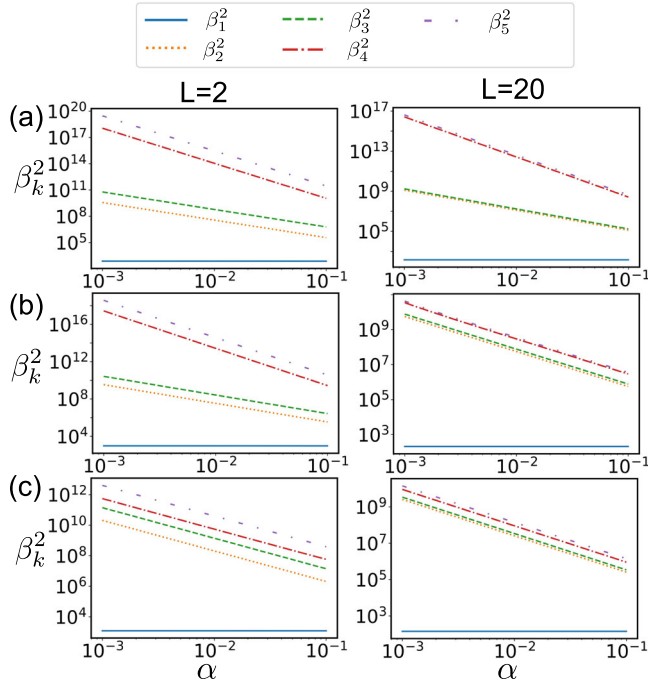

**Fig. 3 | Scaling behavior of $\beta_k^2$.** Scaling of the $\beta_k^2$ as a function of $\alpha$ for imaging two compact sources with distance $L/2$. We consider three different cases: (**a**) direct imaging (**b**) separate SPADE method (**c**) orthogonalized SPADE method. Width of PSF $\sigma = 1$.

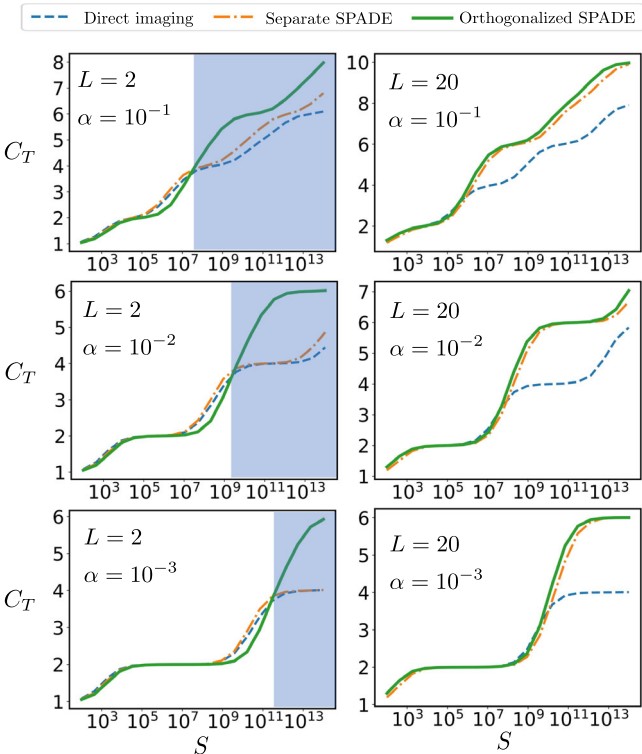

**Fig. 4 | Behavior of total REC for multiple compact sources.** Total REC $C_T$ for direct imaging, the separate SPADE method and the orthogonalized SPADE method as a function of $S$ for imaging of two compact sources with $\alpha = 10^{-1}, 10^{-2}, 10^{-3}$, number of compact source $Q = 2$, $\sigma = 1$. The distance between the centroid of the two compact sources is $L/2$. When $L = 2$, the orthogonalized SPADE method demonstrates a clear advantage over both the separate SPADE method and direct imaging in the shaded region. When $L = 20$, the performance of SPADE and orthogonalized SPADE is comparable (essentially because they become equivalent when the two compact sources are sufficiently far apart) and both outperform direct imaging. Overall, the orthogonalized SPADE protocol demonstrates excellent performance, achieving a high $C_T$ (compared to the best from direct and SPADE protocols) for various choices of $L = 2$ and $L = 20$, as well as across a wide range of sample sizes $S$.

$\Theta(\alpha^n)$ order, lower-order terms must vanish in the probability distribution, particularly those contributions from nearby compact sources. We refer to this new approach as the orthogonalized SPADE method, as it projects onto a basis that is an orthogonalization of the separate SPADE method. This construction applies analogously to any PSF beyond Gaussian PSF. Note that for a single compact source, the separate SPADE and orthogonalized SPADE methods are identical, both referred to as the SPADE method. We find that for the orthogonalized SPADE method

$$\beta_0^2 = 0, \quad \beta_{1 \le i \le Q-1}^2 = \Theta(1),$$
$$\beta_{Q \le i \le 3Q-1}^2 = \Theta(\alpha^{-2}), \quad \beta_{3Q \le i \le 5Q-1}^2 = \Theta(\alpha^{-4}), \cdots \quad (14)$$

The scaling is numerically confirmed in Fig. 3 for the first six eigenvalues, and we expect $\beta_k^2 = \Theta(\alpha^{-2\lceil \lfloor k/Q \rfloor / 2 \rceil})$ to hold for eigenvalues with higher indices and any prior $p(\vec{x})$. In the numerical calculation, we

again consider two compact sources with centroids at $-L/4$ and $L/4$ and a random prior obtained by randomly generating a set of images. We examine the separate SPADE method in Fig. 3(b) and the orthogonalized SPADE method in Fig. 3(c). When the sources are well separated ($L \gg \sigma$), both methods achieve the expected scaling, with four $\beta_k^2$ terms scaling as $\Theta(\alpha^{-2})$, compared to two for direct imaging. The doubling of the number of eigenvalues scaling as $\Theta(\alpha^{-2n})$ for each $n$ aligns with expectations for two compact sources. However, when the sources are closer ($L = 2$, $\sigma = 1$), the performance of the separate SPADE method is strongly degraded, reducing the scaling to that of direct imaging, with only two $\beta_k^2$ scaling as $\Theta(\alpha^{-2})$. In contrast, our orthogonalized SPADE method retains four eigenvalues $\beta_k^2$ with scaling $\Theta(\alpha^{-2})$.

In Fig. 4, we demonstrate the role of $\beta_k^2$ as the thresholds for stepwise increases in total REC $C_T$ for two compact sources. When $L = 2$ (sources close together), direct imaging and separate SPADE show that $C_T$ increases by 2 at each step after the initial two $\Theta(1)$ eigenvalues. In contrast, for orthogonalized SPADE, $C_T$ is increased by 4 after the initial two $\Theta(1)$ eigenvalues, highlighting the advantage of our method for two close compact sources. When $L = 20$ (with sources well separated), both separate SPADE and orthogonalized SPADE yield a $C_T$ increase of 4 at each step. Note that in certain regions of the sample number when $\alpha = 10^{-2}$, the orthogonalized SPADE method may perform slightly worse than the separate SPADE method. This difference arises from the different constant prefactors

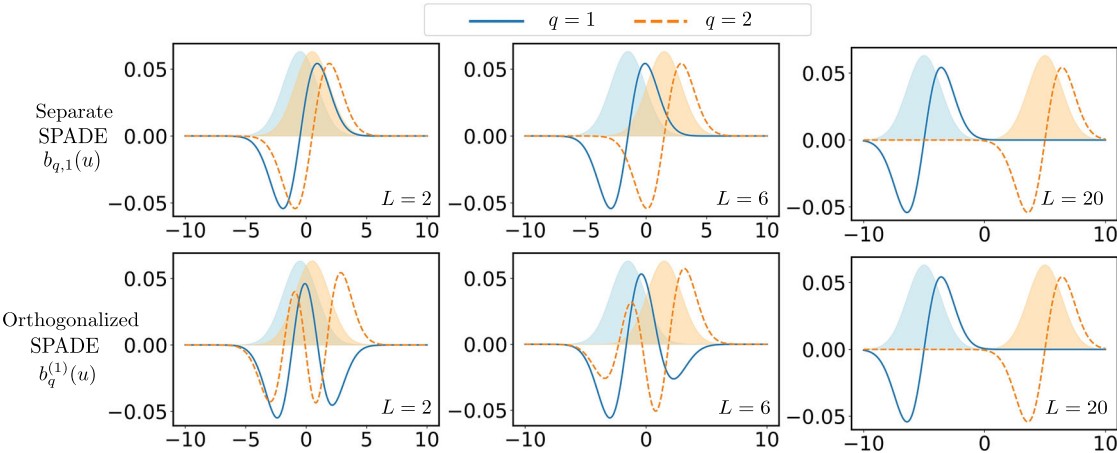

**Fig. 5 | Basis states for the orthogonalized SPADE measurement.** We compare the basis states constructed in Eq. (12) for the orthogonalized SPADE measurement, $|b_q^{(l=1)}\rangle = \int du\, b_q^{(1)}(u)|u\rangle$, with the corresponding basis states for the separate SPADE measurement, $|b_{q,1}\rangle = \int du\, b_{q,1}(u)|u\rangle$. Here, $q = 1, 2$ corresponds to the case of two compact sources ($Q = 2$) with centroids located at $\pm L/4$, so that the separation between the two sources is $L/2$. We examine cases where $L = 2, 6, 20$. The shaded regions show two Gaussian PSFs of width $\sigma = 1$ located at the centers of two compact sources, that are used to construct the basis states $|b_q^{(l)}\rangle$ and $|b_{q,m}\rangle$.

in $\beta_k^2$. To ensure optimal performance, we can adopt an adaptive approach: for a given sample size, we select either the orthogonalized or separate SPADE method based on the total REC of each method, choosing the one that offers superior performance.

Note that in the imaging of multiple compact sources, the eigentasks identified from the eigenvectors $r_k$ generally do not have the simple form seen in the single compact source case. The structure of the eigentasks is strongly influenced by the practical imaging model and the prior knowledge, such as the positions of the individual sources. We explicitly demonstrate this in Supplementary Note 5 E. This observation suggests that, in practical applications, the quantum learning approach offers nontrivial guidance on which features should be incorporated into the downstream analysis.

In Fig. 5, we numerically illustrate the shape of the constructed basis for the separate SPADE method and the orthogonalized SPADE method, considering different distances between the centroids of the two compact sources. Note that the construction of the basis states defined in Eq. (12) does not depend on the size of each source $L_q$. It is evident that when the two compact sources are close to each other, the basis for the separate and orthogonalized SPADE methods differ significantly. However, when the two compact sources are sufficiently far from each other, the basis for the separate and orthogonalized SPADE methods become nearly identical. Given the complicated form of the basis, the spatial light modulator could serve as a practical tool for its implementation, as previously discussed in the context of superresolution[41]. In Supplementary Note 5, we demonstrate that the Hermite-Gaussian mode sorter[42–46] can be used to implement the orthogonalized SPADE method with some additional steps for a Gaussian PSF, and we also provide more details on the derivation of the scaling of $\beta_k^2$ and the eigenvectors $r_k$ based on numerical calculations.

**Demonstrative example**
We now present a demonstrative example to illustrate the advantage of the quantum learning approach and our orthogonalized SPADE method in a face-recognition imaging task. The face images used in our simulation are taken from the Olivetti Faces dataset provided by AT&T Laboratories Cambridge and distributed through `scikit-learn`, with examples shown in Fig. 6b. Each individual has $64 \times 64$ grayscale images with different facial expressions and small variations in pose. These images are converted into one-dimensional images by rasterization, a standard preprocessing technique in machine learning, and the resulting images are partitioned into $M = 3$ segments that serve as compact sources placed over the interval $[-\mathbb{L}/2, \mathbb{L}/2]$ according to the

position function $\zeta(u)$ in Fig. 6a. Our goal is to determine the identity of each given face image. In the following, we treat the imaging system itself as PNNs—the quantum learning devices—and use it to perform this face-recognition task. We emphasize that this approach can address a wide range of imaging tasks beyond this one, such as regression and clustering.

We use face images from $N_{\text{person}} = 20$ individuals, which form the training set whose statistics are used to compute the prior information. Using the REC formalism in Eq. (4), we then calculate the eigenvectors $r_k$. For the $m$th image, the measurement yields a probability distribution $P_m(j)$, where $j$ labels the measurement outcome. The $k$th eigentask is obtained as a linear combination of these distributions with coefficients given by the $k$th eigenvector $r_k$, yielding $\xi_{km} = \sum_j r_{kj} P_m(j)$. This defines the eigentask vector for the $m$th image $\vec{\xi}_m = [\xi_{0m}, \xi_{1m}, \xi_{2m}, \ldots, \xi_{\mathcal{K}m}]$, where the eigentasks are truncated at order $\mathcal{K}$. The truncation is introduced because higher-order eigentasks are noisy and poorly estimated, and including them in downstream analysis, namely training and inference, can degrade the performance. The shapes of the eigenvectors are generally complex, reflecting the complexity of the imaging task, which is common in practical applications. PNNs perform classification by training only the output weights using logistic regression. For each individual, we obtain eigentask vectors $\vec{\xi}_m$, forming datasets that serve as the training inputs for the logistic regression classifier. Before training, each component of $\vec{\xi}_m$ is normalized by dividing by its mean absolute value across the training set to ensure balanced feature scaling. During training, we use logistic regression from the standard Python package `scikit-learn` to fit a multi-class classifier to the labeled eigentask vectors, modeling the probabilities $P(y|\vec{\xi})$ with a multinomial logistic (softmax) model. During testing, we use the new face images for each individual. For each image, we fix the total number of detected photons (sample number) to be $S$. By counting the number of getting each outcome, we obtain an empirical eigentask vector $\hat{\vec{\xi}}_m = [\hat{\xi}_{0m}, \hat{\xi}_{1m}, \hat{\xi}_{2m}, \ldots, \hat{\xi}_{\mathcal{K}m}]$ for the $m$th face image. These empirical eigentask vectors are then used for inference with the trained model.

In the context of imaging multiple compact sources, we show the performance of face recognition using the three approaches in Fig. 6(d1)-(d3). We observe that the success probability $P_{\text{succ}}$ first increases with $\mathcal{K}$ and then decreases. This behavior is intuitive: increasing $\mathcal{K}$ captures more information and improves classification, but beyond a point, higher-order tasks become noisy due to limited sample size $S$, degrading performance and reducing success

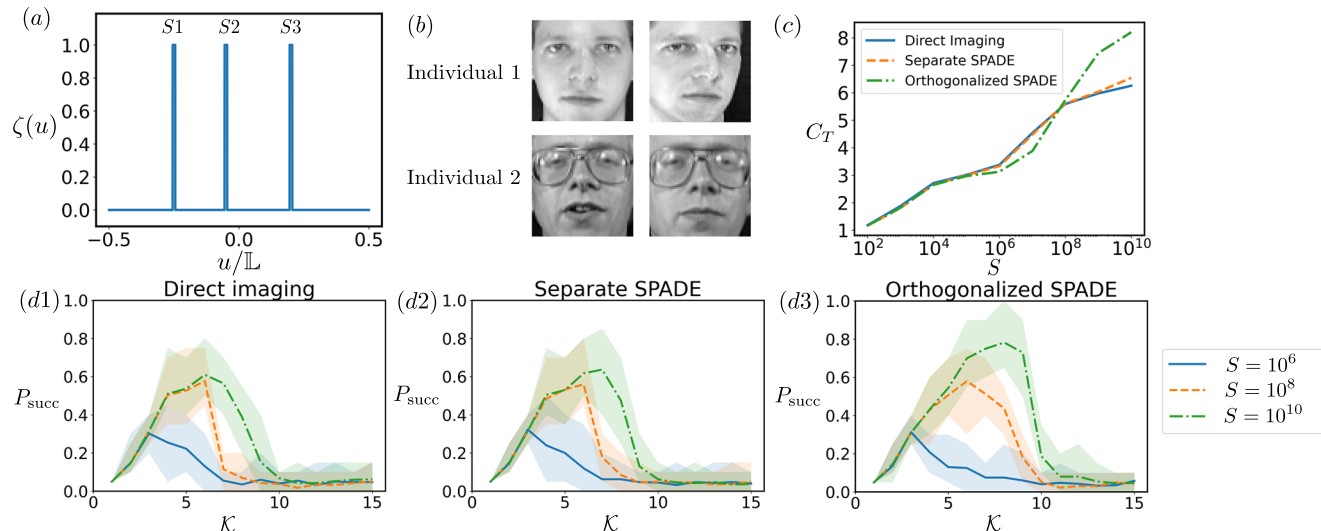

**Fig. 6 | Simulation for face recognition. a** The position function $\zeta(u)$ specifies the positions of three compact sources, labeled $S1$, $S2$, and $S3$. All three compact sources are confined within the interval $[-\mathbb{L}/2, \mathbb{L}/2]$ with $\mathbb{L} = 10$, and each source has a size of 0.1, which is smaller than the PSF width $\sigma = 1$. **b** Example images of two individuals from the Olivetti Faces dataset. **c** The total REC $C_T$ as a function of the sample number $S$ for the three approaches. (d1)-(d3) The success probability $P_{\text{succ}}$

as a function of the highest order $\mathcal{K}$ of the eigentasks included in the training and testing procedures for face recognition, evaluated at different sample sizes $S$. The lines represent the mean success probability, and the shaded region shows the maximum and minimum values across the 20 repetitions. We set $\alpha = 0.1$ in this figure.

probability. Comparing the success probability $P_{\text{succ}}$ of the three approaches, since the source distance is comparable to the PSF width $\sigma$, separate SPADE and direct imaging perform similarly, whereas our orthogonalized SPADE achieves higher performance, with its peak $P_{\text{succ}}$ exceeding the others at $S = 10^{10}$.

This simulation highlights the operational meaning of the total REC $C_T$. First, it estimates the number of eigentasks that can be reliably included in the downstream logistic regression. Since $\beta_k^2$ sets the sample threshold for estimating the $k$th eigentask and $C_T = \sum_k 1/(1 + \beta_k^2/S)$, with each term approaching 1 when $S \gg \beta_k^2$, $C_T$ roughly counts the reliably estimated eigentasks. At $S = 10^6$ and $S = 10^8$, $C_T$ is similar across all approaches, matching the $P_{\text{succ}}$ behavior. At $S = 10^{10}$, $C_T$ for orthogonalized SPADE rises to about 8, aligning with its peak $P_{\text{succ}}$, while direct imaging and separate SPADE reach about 6, consistent with their peaks. Second, a larger $C_T$ allows more well-estimated eigentasks to be included, capturing more information and thus improving the success probability in this example; therefore, a larger $C_T$ indicates better imaging performance. Further details of this simulation, as well as additional examples beyond face recognition, are provided in Supplementary Note 8.

In the broader context, we here present an example that directly addresses the imaging task using PNNs. PNNs employ analog systems with a fixed internal structure—here, the imaging system—and performs practical tasks by training only the output weights, which in this example corresponds to logistic regression, though alternatives like linear regression can be used depending on the task. In this sense, the PNNs framework provides a systematic approach to modeling and solving complex imaging problems, which is especially valuable given the infinite degrees of freedom inherent to imaging. The REC formalism, developed for the PNNs paradigm, further guides the training step by identifying eigentasks and selecting those that are well-estimated and low-noise. This formalism has also proven highly effective in superresolution problems involving complex source structures, as demonstrated in our simulation.

Many machine-learning tasks, including face recognition, can be viewed as discrimination problems that are in principle solvable by the likelihood-ratio method and whose performance is bounded by the Chernoff bound. While simple discrimination settings have been

analyzed using the Chernoff bound in superresolution[22–26], the likelihood-ratio method requires an accurate statistical model of the objects being imaged, whereas learning methods can handle far more complex structures. In our face-recognition example involving tens of individuals, it is infeasible to write down a closed-form likelihood function for an exact likelihood-ratio calculation. Moreover, the Chernoff bound is asymptotic and may appear to be reached even when the success probability is already above 99% in some cases, offering little guidance on the practically relevant regime where we care about reaching moderate performance levels such as 70%. It also does not suggest which measurement strategy should be used when the likelihood-ratio method is inapplicable. By contrast, our learning-based approach yields the total REC as a meaningful figure of merit and, crucially, provides a principled strategy for tackling discrimination tasks in the finite-sample regime. Simulated examples illustrating the above discussion are provided in Supplementary Note 9.

### Imaging general sources beyond the Rayleigh limit

An intriguing question is whether, when a general source cannot be split into multiple compact sources, we can still split the source into $Q$ small pieces and apply our orthogonalized SPADE method, improving the imaging performance. Unfortunately, there is a constraint. Each source generates a set of states $\{|\psi_q^{(m)}\rangle\}_{m=0,1,2,\dots}$, which are used to construct $|b_j^{(l)}\rangle$ via the Gram-Schmidt procedure. When $Q \gg L/\sigma$, the differences between $|\psi_q^{(m)}\rangle$ for different $q$ can be vanishingly small, leading to the potentially suboptimal performance of orthogonalized SPADE method due to large prefactors of the eigenvalues $\beta_k^2$. In Supplementary Note 5C, we demonstrate that when $Q$ exceeds roughly $L/\sigma$, the stepwise increase in $C_T$ is smoothed out, and numerically, we find that both the separate SPADE and the orthogonalized SPADE no longer offer advantages over direct imaging. In conclusion, the superresolution methods from refs. [1,7] can be properly generalized to resolve multiple compact sources but may fail for generally distributed sources. We also present a discussion of imaging such a general source using direct imaging, showing that $C_T$ is approximately related to the ratio between the source size and the PSF width, as detailed in Supplementary Note 4.

## Discussion

In this work, we treat the imaging systems in superresolution as PNNs, i.e., quantum learning devices, thereby providing a systematic framework for addressing practical imaging tasks with complex structures and operating in the finite-sample regime. Based on the REC formalism, the measurable features of the source and their corresponding sample thresholds can be identified with eigentasks, while the total REC serves as a principled metric for selecting relevant features for downstream analysis and quantifying the performance of an imaging method. We further extend the superresolution framework to handle multiple compact sources and propose the orthogonalized SPADE method−a nontrivial generalization that relaxes the strong assumptions of earlier superresolution studies, thereby improving practical applicability. We show that superresolution exhibits a stepwise increase in total REC, with thresholds determined by the ratio of source size to PSF width. The advantages of this quantum learning approach and our orthogonalized SPADE method are demonstrated through total REC calculations and concrete examples, including face recognition.

It would also be worthwhile to investigate other potentially advantageous imaging protocols, e.g., entangled measurements on multiple copies of photon states, which exhibit advantages over separable measurements in tomography[47,48]. We may also explore the potential applications of quantum computing schemes[49–51], where the quantum advantage inherent in these schemes could benefit specific imaging tasks. Note that the total REC depends on the POVM used in the detection. It may be interesting to explore whether a closed-form expression of the total REC optimized over all possible POVMs can be found, at least in some special cases−for example, when the measurable features require compatible measurements.

## Methods

### Reparameterization invariance of total REC and eigentasks

We emphasize that a key advantage of the REC formalism is that the identified eigentasks and the total REC are invariant under reparameterization. These eigentasks are determined by the prior information, the physical model of the learning device, and the structure of the learning tasks. They are then used as the actual feature vectors for downstream analysis, including model training and inference with the trained model. This ensures that the results are not artificially influenced by the choice of parameterization. The invariance under reparameterization is formally established in the following proposition.

**Proposition 1.** Let $\rho(\boldsymbol{\theta})$ be a family of states parameterized by $\boldsymbol{\theta}$ with prior $p(\boldsymbol{\theta})$, a fixed POVM $\{M_i\}_{i=0}^{K-1}$, sample size $S$, and measured features $\eta_i(\boldsymbol{\theta}) = Tr[\rho(\boldsymbol{\theta})M_i]$. In the parameterization of $\boldsymbol{\theta}$, the total REC at sample size $S$ is $C_T(S)$, with eigenvalues $\{\beta_k^2\}$ and eigentasks defined by solving Eq. (4), yielding $f_k(\boldsymbol{\theta}) = \sum_j r_{kj}\eta_j(\boldsymbol{\theta})$. Let $\boldsymbol{\phi} = h(\boldsymbol{\theta})$ be a bijective differentiable reparameterization, with pushforward prior $p_\Phi(\boldsymbol{\phi}) = p(h^{-1}(\boldsymbol{\phi}))\,|\det J_{h^{-1}}(\boldsymbol{\phi})|$ so that $p_\Phi(\boldsymbol{\phi})\,d\boldsymbol{\phi} = p(\boldsymbol{\theta})\,d\boldsymbol{\theta}$, and reparameterized family $\widetilde{\rho}(\boldsymbol{\phi}) = \rho(h^{-1}(\boldsymbol{\phi}))$. As in Eq. (4), form $\widetilde{\rho}^{(t)}$, $\widetilde{D}$, $\widetilde{G}$, $\widetilde{V}$, the generalized spectrum $\{\widetilde{\beta}_k^2\}$, eigentasks $\widetilde{f}_k(\boldsymbol{\phi}) = \sum_j \widetilde{r}_{kj}\,\widetilde{\eta}_j(\boldsymbol{\phi})$ with $\widetilde{\eta}_j(\boldsymbol{\phi}) = \eta_j(h^{-1}(\boldsymbol{\phi})) = \eta_j(\boldsymbol{\theta}) = Tr[\rho(\boldsymbol{\theta})M_j]$, and total REC $\widetilde{C}_T(S)$. Then:

1. (Total REC invariance)

$$\widetilde{C}_T(S) = C_T(S).$$

2. (Spectral invariance)

$$\{\widetilde{\beta}_k^2\} = \{\beta_k^2\}.$$

3. (Eigentask invariance)

$$\widetilde{f}_k(\boldsymbol{\phi}) = f_k(h^{-1}(\boldsymbol{\phi})) = f_k(\boldsymbol{\theta}).$$

**Proof.** By definition and the pushforward prior,

$$
\begin{aligned}
\widetilde{\rho}^{(t)} &= \int_\Phi \widetilde{\rho}(\boldsymbol{\phi})^{\otimes t}\, p_\Phi(\boldsymbol{\phi})\, d\boldsymbol{\phi} \\
&= \int_\Phi \rho(h^{-1}(\boldsymbol{\phi}))^{\otimes t}\, p(h^{-1}(\boldsymbol{\phi}))\,|\det J_{h^{-1}}(\boldsymbol{\phi})|\, d\boldsymbol{\phi}.
\end{aligned}
\tag{15}
$$

Set $\boldsymbol{\theta} = h^{-1}(\boldsymbol{\phi})$. Then $d\boldsymbol{\phi} = |\det J_h(\boldsymbol{\theta})|\, d\boldsymbol{\theta}$ and $|\det J_{h^{-1}}(\boldsymbol{\phi})| = 1/|\det J_h(\boldsymbol{\theta})|$, so the Jacobians cancel:

$$\widetilde{\rho}^{(t)} = \int_\Theta \rho(\boldsymbol{\theta})^{\otimes t}\, p(\boldsymbol{\theta})\, d\boldsymbol{\theta} = \rho^{(t)}\ (t=1,2).$$

This relation implies

$$
\begin{aligned}
\widetilde{D}_{ii} &= Tr[M_i\,\widetilde{\rho}^{(1)}] = Tr[M_i\,\rho^{(1)}] = D_{ii}, \\
\widetilde{G}_{ij} &= Tr[(M_i \otimes M_j)\,\widetilde{\rho}^{(2)}] = Tr[(M_i \otimes M_j)\,\rho^{(2)}] = G_{ij}.
\end{aligned}
\tag{16}
$$

Hence $\widetilde{V} = \widetilde{D} - \widetilde{G} = D - G = V$. With $\widetilde{G} = G$ and $\widetilde{V} = V$, we obviously have

$$
\begin{aligned}
\widetilde{C}_T(S) &= Tr((\widetilde{G} + \widetilde{V}/S)^{-1}\widetilde{G}) \\
&= Tr((G + V/S)^{-1}G) = C_T(S).
\end{aligned}
\tag{17}
$$

The generalized eigenproblems

$$V r_k = \beta_k^2\, G r_k, \quad \widetilde{V}\,\widetilde{r}_k = \widetilde{\beta}_k^2\, \widetilde{G}\,\widetilde{r}_k,$$

are also identical, so $\{\widetilde{\beta}_k^2\} = \{\beta_k^2\}$ and $\widetilde{r}_k = r_k$. Using $\widetilde{\eta}(\boldsymbol{\phi}) = \eta(h^{-1}(\boldsymbol{\phi}))$,

$$
\begin{aligned}
\widetilde{f}_k(\boldsymbol{\phi}) &= \sum_j r_{kj}\,\widetilde{\eta}_j(\boldsymbol{\phi}) = \sum_j r_{kj}\,\eta_j(h^{-1}(\boldsymbol{\phi})) \\
&= f_k(h^{-1}(\boldsymbol{\phi})) = f_k(\boldsymbol{\theta}).
\end{aligned}
\tag{18}
$$

## Data availability

No data sets were generated or analyzed during the current study.

## Code availability

All codes used in this paper have been deposited in GitHub at https://github.com/ykwang-phys/quantum-learning-imaging.

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

## Acknowledgements

We would like to thank Hakan E. Tureci for helpful discussion. Y.W. and S.Z. acknowledge funding provided by Perimeter Institute for Theoretical Physics, a research institute supported in part by the Government of Canada through the Department of Innovation, Science and Economic Development Canada and by the Province of Ontario through the Ministry of Colleges and Universities. Y.W. also acknowledges funding from the Canada First Research Excellence Fund. L.J. acknowledges support from the ARO(W911NF-23-1-0077), ARO MURI (W911NF-21-1-0325), AFOSR MURI (FA9550-19-1-0399, FA9550-21-1-0209, FA9550-23-1-0338), DARPA (HR0011-24-9-0359, HR0011-24-9-0361), NSF (OMA-1936118, ERC-1941583, OMA-2137642, OSI-2326767, CCF-2312755), NTT Research, Packard Foundation (2020-71479), and the Marshall and Arlene Bennett Family Research Program. J.L. is supported in part by the

University of Pittsburgh, School of Computing and Information, Department of Computer Science, Pitt Cyber, Pitt Momentum fund, PQI Community Collaboration Awards, John C. Mascaro Faculty Scholar in Sustainability, Thinking Machines Lab, Cisco Research, funding from IBM Quantum through the Chicago Quantum Exchange, and AFOSR MURI (FA9550-21-1-0209). C.O. was supported by the NRF Grants (No. RS-2024-00431768 and No. RS-2025-00515456) funded by the Korean government (MSIT) and IITP grants funded by the Korean government (MSIT) (No. IITP-2025-RS-2025-02283189 and IITP-2025-RS-2025-02263264).

## Author contributions

Y.W. carried out the analytical calculation and the numerical simulation. L.J. conceived the project. S.Z., J.L., and L.J. supervised the project. Y.W., C.O., J.L., L.J., and S.Z. contributed to the development of ideas and the writing of the manuscript.

## Competing interests

The authors declare no competing interests.
