## [Transparent Peer Review file · Nature Communications]

Advancing quantum imaging through learning theory

Corresponding Author: Dr Yunkai Wang

Version 0:

Reviewer comments:

Reviewer #1

(Remarks to the Author)

The manuscript by Wang et al. introduces a novel framework for quantifying imaging performance using the Resolvable Expressive Capacity (REC), drawing on recent advances in learning theory. They apply the formalism to several imaging scenarios, including two-point resolution, single and multiple sources. REC provides an interesting alternative to Fisher information.

While this work may have combined learning and imaging in a novel way, I cannot make a recommendation at this point for the following reasons.

The writing style of the manuscript is overly technical and jargon-heavy, which is likely inaccessible to the broad readership of Nature communications. For this venue, the authors should significantly revise the presentation to emphasize conceptual clarity and physical intuition, ensuring that readers can appreciate the work outside the immediate quantum information community who are already familiar with the work of Tsang and etc.

Important ideas are not always clearly explained before the mathematics is introduced, and the text is frequently interrupted by dense equations. This makes the overall structure feel disjointed, and it can be difficult for the reader to understand the main points. Between page 1 and page 3, I counted at least 20 different variables. The paper would benefit from clearer explanations, better organization, and a more intuitive presentation of the key concepts.

It would be helpful for the senior authors to step in and help improve the narrative, clarify the motivation, and guide the presentation toward a more accessible and coherent presentation.

The clarity and flow of each individual sentence could be improved. For example, the first one in the introduction could be :

"The quality of an image formed by a single-lens system depends on several factors, including the lens's resolution, the measurement strategy employed in the image plane, and the number of collected samples."

The following needs to be more mathematically/rigorously defined:

* W in Eq.1 in the minimisation

* What value does Eq. 1 take, and what it means in the limit that $C[f]$ approaches either limits.

* Second column, page 2: "This discussion advances the superresolution to a more practical scenario, where the focus shifts from a single compact source within the Rayleigh limit to multiple compact sources". This sentence makes no sense.

The definition of a "compact source" is not given until page 3, yet is it mentioned several times prior.

* The sentence on lines 71-76: The total REC: [...] serves as a single parameter quantification of imaging quality with a well-motivated interpretation...[...]. The expression followed is not a clear interpretation.

* The presentation in Fig 1 (b) appear counter intuitive: why does the total REC appear to be higher for smaller features? Q , β_0 and β_k in the caption of Figure 1 are not defined at this point

* Page 3: the cluster of equations between 2-6 are not well-explained.

* Page 3, line 29 refers to this work as a 'letter'; same goes for page 3 of the supplemental material, line 12.

I am unable to follow much beyond page 3: from page 3 onwards as the manuscript appear to be a discussion of dense symbols that are not very well-discussed.

Supplemental material:

$l(u)$ in Eq 1 is not defined.

Eq: 12, a more intuitive/physical interpretation of the different beta needs to be given

(Remarks on code availability)

Reviewer #2

(Remarks to the Author)

The authors discuss an important application of quantum learning theory to the problem of superresolution imaging. By combining analytical approximations with numerical computations and simulations, they assess the impact of finiteness of the number of samples drawn from the probability distribution of specific POVMs to estimate the parameters of a one-dimensional (1D) optical source, specifically their spatial moments, from such data samples. Their choice of POVMs generalizes those proposed in two of the authors' earlier work (Ref. 5), but instead of Fisher information here they use the notion of resolvable expressive capacity (REC), introduced previously in Ref. 37, to quantify the number of degrees of freedom that can be accurately estimated with increasing sample size. They have employed the REC metric to contrast the generally inferior performance of direct imaging with superior superresolution protocols that employ wavefront projections, particularly in the sub-Rayleigh regime of source brightness features. They have introduced a new concept of orthogonalized SPADE basis that is optimized very generally, including the case of multiple compact sources that are in close proximity of one another regardless of each source's size and their mutual separation in relation to the Rayleigh resolution scale.

Here are some major observations that the authors must consider in improving the paper from its current form:

1. While the general results are presented with clarity, their over-reliance on and over-referencing of derivations presented in a large Supplemental document sometimes tends to obscure how these results connect with one another. I understand the need for brevity in the main paper, but I believe clarity can be improved in this respect with the inclusion of only a little extra text in places. An example of this kind of "disconnect" is between Eqs. (1) and (2)-(5), which are separated by a lot of text. I recommend that the authors bring forward the latter set of equations ahead of the paragraph entitled "Resolving two-point sources." In fact, a brief review of the REC concept covered in these equations with emphasis on the physical significance and origin of the various matrices in a separate section, before its applications to the different problem scenarios are discussed, will be much appreciated by the general reader.

Another instance of a lack of continuity is a discussion of Fig. 3 before the actual twin source structure to which the figure applies is first introduced in the following paragraph (which starts as "We perform a numerical..."). The order of these two discussions needs to be flipped.

2. Most of the discussion of the paper is centered on scaling laws for the squared eigenvalues β_k^2 that govern the fidelity of estimation of the various parameter combinations. Apart from the case of a symmetric point-source pair, no specific, quantitative statements about the prefactors associated with these scaling laws are made, however. The numerical results that are presented in a number of figures certainly contain these prefactors numerically, but little attempt has been made to clarify their general form or dependences on the various source parameters. Their relative strengths affect the C_T curves, e.g., in making direct imaging perform better than SPADE in some ranges of the sample size, S , as seen in Fig. 4. This is a serious shortcoming of the presentation, but one which can be remedied by presenting some further discussion of these prefactors, even if it is only at the level of their numerically extracted values.

3. It is also important to stress in the paper that the total REC is independent of the choice of the complete basis of the f -functions, and only depends on the physical system and the POVM used to acquire the data. In this connection, it may also be worth pointing out if attempts have been made, perhaps by others, to maximize the total REC over all possible POVMs to give it a truly invariant property of a given parametrized quantum state.

4. I also do not see how the authors can assert that for SPADE-based superresolution, the n th estimated function, $\sum r_{\{nm\}} P_m$, only involves the $2n$ th and $(2n+1)$ th moments in the limit $\alpha \rightarrow 0$. For this to be true, the matrix of eigenvectors $r_{\{nm\}}$ has to be nearly diagonal, up to permutations, in the limit $\alpha \rightarrow 0$. Otherwise, each estimated function would be a linear combination of many moments that is dominated by the lower-order moments in that limit. This point needs clarification in the second half of the paragraph containing Eq. (8), since the parallel with direct imaging discussed in the previous paragraph is unclear.

5. The presence of a slowly increasing plateau in total REC (Fig. 2(c)) as $S \rightarrow \infty$ for compact sources that are large compared to the PSF width needs further examination, perhaps by means of an asymptotic approximation of the total REC in this limit. I believe that the plateau would never stop rising with increasing S , but its increase would continue to maintain a logarithmic type of dependence on S with a relatively small coefficient. Indeed, in the limit of infinitely many photons even direct imaging in the absence of noise should be able to superresolve any compact source to an arbitrary degree of fineness - and thus attain arbitrarily large values of C_T - regardless of the PSF width, as a number of investigations over the last fifty years have shown (see, e.g., for superresolving a point-source pair, Lucy, L., "Statistical Limits to Super Resolution," *Astron. Astrophys.*, vol.261, 706-710 (1992)).

A number of minor errors and comments are listed below.

1. Page 1, line 72, over what space is the set of functions $\{f_l\}_l$ orthonormal and complete? I imagine it is the space over which the system parameters $\langle \theta \rangle$ live, but this should be simply stated.
2. Line 49, page 2, the Gaussian function, $\phi_0(u)$, has the power of 2 missing from u inside the exponent.
3. Line 92, page 3, the relation $\beta_{k^2} = \Theta(\alpha^{2 \lceil k/2 \rceil})$ has a negative sign missing in the exponent.
4. Line 26, page 5, a similar relation has not just the minus sign missing in the exponent but, I believe, the $\lceil k/Q \rceil$ should be replaced by $\lfloor k/Q \rfloor$ so this relation becomes consistent with Eq. (11).
5. Line 56, page 5, the expression for β_{k^2} is incorrect. I believe it should be $\beta_{k^2} = \Theta(\alpha^{-2 \lceil \lfloor k/Q \rfloor / 2 \rceil})$ to be consistent with Eq. (14).
6. Two instances of power 2 are missing in the last two expressions giving approximations for S in the caption of Fig. 1.
7. In Fig. 4, the dotted and dashed curves are hard to see clearly in some of the plots. I suggest making them thicker and/or reducing the thickness of the solid curves.
8. Line 33, page 7, replace "a issue that are" by "an issue that is." Similar grammatical errors occur elsewhere too. Authors, please look for them.
9. Line 64, page 3, change "providing" to "provides."

The paper discusses an important topic from the new perspective of sample-based capacity of quantum measurements on a system state to extract certain linear combinations of the state parameters accurately. If the authors can sufficiently address all of the points I have raised here regarding the current version of the manuscript, then I can recommend its publication in *Nature Communications*.

(Remarks on code availability)

Reviewer #3

(Remarks to the Author)

The authors present a learning theory approach to the expanding field of quantum-inspired superresolution. In particular, they consider the resolvable expressive capacity (REC) as an image quality quantifier. The total REC represents the total number of linearly independent functions that can be expressed as a linear combination of the measurement results. This quantity presents some interesting properties that allow to identify how the ability to learn sub-diffraction features of an object depend on the sample size. In particular, one can observe jumps in the dependence of the total REC with the sample size, and the amplitude of these jump depend on the kind of measurement that is performed.

I find interesting how this result agrees with the prediction of quantum-inspired superresolution theory that spatial mode demultiplexing (SPADE) techniques can overcome the resolution of direct imaging. I also find theoretically interesting how one can use the total REC to gain an intuition on how much one can divide an extended source in terms of subdiffraction objects to be analyzed by the appropriate orthogonalized SPADE techniques. However, I have a hard time to understand

the operational meaning of these results. In other words, how can one relate the REC with a procedure to reconstruct an image of the object with super-resolution features? Without an answer to this question, I cannot recommend publication in Nature Communications.

I also think that the bibliography is a bit outdated, and there several technical details that should be clarified before resubmitting to Nature Communication or elsewhere. I will detail these, point by point in the following.

- The authors identify to main advantages of the REC over the quantum Fisher information matrix (QFIM): 1) the REC is a scalar quantity with a clear interpretation, while to reduce the QFIM to a scalar one needs to introduce a weight matrix 2) the QFIM is relevant in the asymptotic regime, while the REC is valid also in the finite sample regime.
Regarding 1): the QFIM gives a clear and complete (up to incompatibility issues) picture of a multiparameter estimation problem, providing a lower bound on the parameters' covariance matrix. Most importantly, it gives a bound on a precise image reconstruction method which consists in estimating a given number of parameters fully defining the observed object. If one needs a scalar metric one can use a weight matrix, but in principle one has the full information in the QFIM. On the other hand, which is the learning task that allows to reconstruct the object from the measurement data? What does the REC tells us about how well can we do this? Can the authors reply to these questions? If not, I think that the clear operational meaning of the QFIM makes it a much better quantifier for the performances of an imaging task.
Regarding 2) it is true that the QFIM has a clear interpretation only in the asymptotic regime. However, the authors results show that for the total REC to be non-zero one needs thousands when not millions of samples, so is the asymptotic value of the QFIM a practical limitation in this contest? And if one was really interested in meaningful finite-sample results, why on should prefer a learning theory approach instead of a Bayesian estimation theory approach?
- The bibliography is incomplete and outdated. In particular, the most recent experiments using multimode Hermite-Gaussian mode sorters are missing: Rouviere et. al. *Optica* 11 166 (2024) where the record sub-Rayleigh sensitivity in separation estimation has been reached, Tan et al. *Optica* 10 1189 (2023), where the moments estimation approach considered by the authors was implemented, Santamaria et al. *Optica Quantum* 2 46 (2024) where sources of different intensity have been considered.
Other papers that might be interesting for the authors are: Sorelli et al. *Phys. Rev. Lett.* 127 123604 (2021) where it was shown that estimators only based on linear combinations of the first moments of the measurement results can saturate the ultimate quantum limit. This seems to be similar to the authors observation that linear combinations of the measurement probability can be used to reconstruct moments of the object.
The authors distinguish between sub-Rayleigh features and features larger than the Rayleigh limit and discuss how different measurements are effective in the different regimes. They might be interested in the results of Grace et al. *JOSA A* 37 1288 (2020) where it was investigated how to switch between direct imaging and SPADE for the simultaneous estimation of separation and centroid.
- Another point that the authors should clarify is the role of the prior probability in the calculation of the REC. The only point, where the authors explicitly state their prior is in the two sources problem. In this context, they assume that the width of the prior is much smaller than the width of the point spread function of the imaging system. I find this assumption problematic, since it seems to suggest that one needs prior subdiffraction information to achieve subdiffraction resolution. Could the authors clarify if this the case and which assumptions on the prior probability are made in the more general scenarios considered in the paper?
- Commenting Fig. 2 (b) the authors say that the fact that the total REC increases by two instead of one at each step is expected. Do they expect this because of the scaling of the eigenvalues in fig 2 (a), or is there a more intuitive explanation?
- Commenting the results in Fig. 2 the authors say that the saturates at value which is approximately L/σ . But the plots show a value that is systematically larger. Did the authors tried to estimate this value? Also, the plots are generated for specific values of L fixing σ . Did the authors tried to use different values of σ and L , but keeping the ratio fixed to justify their claim that L/σ is the important quantity here?
- The caption of Fig. 5 seems wrong. It suggests that the centroids of the two extended sources are fixed, but the size is changed. But from the plot (and the main text) it seems that the size of the sources is fixed, and their distance is changed. The authors should correct or clarify what they are showing.
- Commenting Fig. 5 in the text, the authors refer to Hermite Gaussian mode sorters and say that in Sec V D of the supplementary they show that these sorters can be used to implement orthogonalized SPADE. I couldn't find this information there. In fact, all the references about the mode sorters are included in the bibliography of the supplementary material but are not cited in the text. Either the statement in the main text should be removed, or the supplementary material should be amended.

(Remarks on code availability)

Version 1:

Reviewer comments:

Reviewer #1

(Remarks to the Author)

The authors have addressed my previous concerns and improved the clarity of the manuscript. I can recommend its publication.

(Remarks on code availability)

The authors have addressed my previous concerns and improved the clarity of the manuscript. I can recommend its publication.

Reviewer #2

(Remarks to the Author)

I have now reviewed both the revised submission and detailed responses of the authors to all reviewers' comments, suggested corrections, and recommendations for the manuscript. The manuscript is now substantially improved in its clarity and rigor of presentation and flow, with the various statements and arguments largely well supported by additional mathematical calculations, figures, improved figure captions, and qualitative statements included either in the manuscript or its supplementary materials. It is unfortunate that the supplementary materials are far more voluminous and detailed than the main paper, which still lends a certain opaqueness to the arguments in the main text, but I do believe this is about the best balance that could be struck without making the main paper too bulky and difficult to follow. The authors have painstakingly addressed all my original review comments in a satisfactory manner, and I therefore recommend the publication of the manuscript in Nature Communications in its revised form.

(Remarks on code availability)

Reviewer #3

(Remarks to the Author)

The manuscript was significantly rewritten and now it is more accessible to readers without a strong background in learning theory, and it is far more readable without continuously looking into the supplementary material.

My main criticism was that the previous version lacked a practical interpretation of the REC results. The authors addressed this point by improving their presentation and adding a "Demonstrative Example" section. This is certainly helpful, however I find that there is a dissonance between the example they choose and the criticism they raise to the QFIM approach.

I agree with the authors that when performing imaging tasks, i.e. reconstructing the shape of the objects, if we want to use estimation theory (i.e. the QFIM matrix approach) we need to choose a parametrization of the object, and that finding the best one is far from trivial. I also agree with the authors that in this context a learning approach that finds by itself the right parametrization is very attractive especially for complex scenes or in experimental situations where it is impossible to fully characterize the system.

On the other hand, the classification task with QR-code-like objects studied by the authors is not an image reconstruction task, but an hypothesis testing task, and as such it would be unnatural to analyze it with estimation theory. There is quite some literature on using SPADE for hypothesis testing tasks, in particular Grace, and Guha, PRL 129, 180502 (2022) proved the optimality of SPADE for arbitrary multiple hypothesis testing tasks, and identified bounds on the error probabilities for both symmetric and asymmetric hypothesis testing scenarios. They also considered the sorting of QR-codes. For such a task likelihood ratio kind of tests are known to achieve optimal results. It would be extremely interesting if the authors could compare the error probabilities achieved with their approach with those ultimate bounds. Can the authors reach the ultimate bounds with their approach? Does their approach need a less precise characterization of the system to perform the testing since the system can learn by itself?

If the authors could answer positively to these questions, and clarify that with little increased complexity their approach could generalize to more complex tasks like face recognition, or image reconstruction, then this work would definitely deserve to be published in nature communication.

If not, I think it would not meet the impact criteria of this journal, and would be more appropriate for a more technical one.

(Remarks on code availability)

Version 2:

Reviewer comments:

Reviewer #3

(Remarks to the Author)

The authors addressed all my concerns. They proved that for simple tasks the learning approach is equivalent to optimal statistical methods. Moreover, the learning approach can be applied to complex tasks where a full statistical model of the problem is difficult if not impossible to formulate, and provide satisfactory results.

The paper in this form can be published in nature communication.

(Remarks on code availability)

We thank the referee for the valuable comments, which have led to a substantial revision of the manuscript. All the points raised have been carefully considered. Our detailed responses are provided below, with the corresponding revisions highlighted in blue in the main text and Supplemental Material, and described in each reply.

Report of the First Referee

Referee's comment:

The manuscript by Wang et al. introduces a novel framework for quantifying imaging performance using the Resolvable Expressive Capacity (REC), drawing on recent advances in learning theory. They apply the formalism to several imaging scenarios, including two-point resolution, single and multiple sources. REC provides an interesting alternative to Fisher information.

While this work may have combined learning and imaging in a novel way, I cannot make a recommendation at this point for the following reasons.

The writing style of the manuscript is overly technical and jargon-heavy, which is likely inaccessible to the broad readership of Nature communications. For this venue, the authors should significantly revise the presentation to emphasize conceptual clarity and physical intuition, ensuring that readers can appreciate the work outside the immediate quantum information community who are already familiar with the work of Tsang and etc.

Important ideas are not always clearly explained before the mathematics is introduced, and the text is frequently interrupted by dense equations. This makes the overall structure feel disjointed, and it can be difficult for the reader to understand the main points. Between page 1 and page 3, I counted at least 20 different variables. The paper would benefit from clearer explanations, better organization, and a more intuitive presentation of the key concepts.

It would be helpful for the senior authors to step in and help improve the narrative, clarify the motivation, and guide the presentation toward a more accessible and coherent presentation.

The clarity and flow of each individual sentence could be improved. For example, the first one in the introduction could be :

"The quality of an image formed by a single-lens system depends on several factors, including the lens's resolution, the measurement strategy employed in the image plane, and the number of collected samples."

Our reply:

We thank Referee 1 for the revision suggestions. We have substantially revised the writing of our work. The main changes are as follows:

1. We added a concrete example in the new section "Demonstrative example" in the main text along with the newly added Sec. VIII in the Supplemental Material, to illustrate the advantage of the quantum learning approach and our orthogonalized SPADE method.

2. We reorganized the introduction and the presentation of the REC formalism to clarify the two main contributions of this work before presenting the detailed calculations: introducing the quantum learning framework to address practical imaging tasks with complex structures, and generalizing earlier superresolution results to broaden the applicability of the approach. Greater emphasis is placed on the physical intuition and underlying motivation behind our work.

3. We improved the clarity of each individual sentence and provided more detailed explanations before introducing symbols and new concepts to ensure smooth transitions between sections. The presentation of

the calculations has also been reorganized to separate different aspects of the analysis, making it easier to follow, enhancing coherence, and reducing reliance on the Supplemental Material.

4. We provide a summary of the notations at the beginning of the Supplemental Material to help readers follow the analysis.

***The following needs to be more mathematically/rigorously defined:
W in Eq.1 in the minimisation***

Our reply:

We have added the following explanation under Eq. 1 of the main text: “ W_j represents the weight coefficient in front of $\bar{P}_j(\theta)$ to be optimized to achieve the optimal linear approximation of $f(\theta)$ ”

What value does Eq. 1 take, and what it means in the limit that $C[f]$ approaches either limits.

Our reply:

$C[f] = 1 - \min_w \frac{\mathbb{E}_\theta \left[\mathbb{E}_X \left[(\sum_j W_j \bar{\eta}_j(\theta) - f(\theta))^2 \right] \right]}{\mathbb{E}_\theta [f(\theta)^2]}$, which takes values between 0 and 1, represents the relative error of approximating the target function $f(\theta)$ as a linear combination of the measured features $\bar{\eta}_j(\theta)$. Here, $\bar{\eta}_j(\theta)$ denotes the measured features, which approximate the actual features $\eta_j(\theta)$. If $f(\theta)$ can be approximated with the actual value of features $\eta_j(\theta)$, we have $\sum_j W_j \bar{\eta}_j(\theta) - f(\theta) \approx 0$. $\bar{\eta}_j(\theta)$ is the measured feature as an approximation of the actual feature $\eta_j(\theta)$, and when the sample number S is not sufficiently large, the deviation of $\bar{\eta}_j(\theta)$ from $\eta_j(\theta)$ will cause some error. But when $S \rightarrow \infty$, we have a very good approximation. In this case, $C[f] \approx 1$. But when we cannot approximate the target function $f(\theta)$ well, we can at least choose $W_j = 0$ for $\forall j$, which means $C[f]$ has minimal value 0.

We try to add more explanation under Eq. 1 of the main text: “REC $C[f]$, which takes values between 0 and 1, can be understood as the normalized mean-squared accuracy of approximating $f(\theta)$, where $C[f] = 1$ represents a perfect approximation using the measured features, and deviation from 1 indicates that the target function cannot be well approximated.”

We also move the later discussion of REC formalism to combine with Eq. 1 of the main text to make the introduction of this concept more accessible to the readers.

We also added more details under Eq. 12 of the Supplemental Material: “ $C[f]$ takes values between 0 and 1, representing the relative error of approximating the target function ... But when $S \rightarrow \infty$, we have a very good approximation. In this case, $C[f] \approx 1$.”

Second column, page 2: "This discussion advances the superresolution to a more practical scenario, where the focus shifts from a single compact source within the Rayleigh limit to multiple compact sources". This sentence makes no sense.

Our reply:

The superresolution discussion prior to our work typically assumes that the entire source has a size much smaller than the Rayleigh limit, which we refer to here as a single compact source. However, in imaging, a typical resolution discussion considers sources whose overall size can exceed the Rayleigh limit,

while the specific fine features of interest lie within the Rayleigh limit. In this sense, restricting attention to a single compact source imposes strong assumptions and limits the practical applicability of the method.

Despite its practical importance, the problem of imaging multiple compact sources—each individually sub-Rayleigh but collectively covering a larger region—has not been systematically explored in the context of superresolution. When nearby sources are spaced at distances comparable to the point spread function (PSF) width, they introduce additional noise, and naive applications of earlier superresolution methods fail to outperform direct imaging. To overcome this limitation, we propose the orthogonalized SPADE method—a new measurement strategy that maintains strong performance in resolving multiple compact sources.

Therefore, we claim that imaging multiple compact sources constitutes a nontrivial generalization of the earlier SPADE method, representing a step toward making the superresolution approach more practical. At the very least, our measurement construction enables us to handle certain general images whose overall size exceeds the Rayleigh limit but whose structures remain somewhat clustered. In this sense, we aim to provide a generalization of the superresolution approach that is more practically useful. We hope that this discussion lays the groundwork for imaging general sources whose overall size exceeds the Rayleigh limit, while still resolving fine details of the source using superresolution techniques.

We have revised the final paragraph of the Introduction to clarify the motivation here: “Besides adopting a quantum learning perspective to study the imaging problem, ... constituting a nontrivial generalization of the earlier SPADE method and a step toward making the approach more practical.”

The definition of a "compact source" is not given until page 3, yet is it mentioned several times prior.

Our reply:

We added the definition for the term compact source in the first paragraph of the introduction: “The concept of superresolution has been extended in various directions [2], focusing on imaging a single compact source—an object much smaller than the Rayleigh limit.”

The sentence on lines 71-76: The total REC: [...] serves as a single parameter quantification of imaging quality with a well-motivatd interpretation...[...]. The expression followed is not a clear interpretation.

Our reply:

We revised the sentence to clarify its meaning and provided interpretations for each expression it contains: “To quantify the overall performance of an imaging system, we are interested in identifying the set of functions $f(\theta)$ that can be approximated in this way and in quantifying the effective size of this set, ... Intuitively, C_T quantifies how many independent features of the underlying signal can be captured by the measurement process, and capturing more features improves performance on complex learning tasks by providing greater freedom to approximate the target function $f(\theta)$.”

A more concrete example that illustrates the meaning of C_T in greater depth is also provided in the new section “Demonstrative example” in the main text along with the newly added Sec. VIII in the Supplemental Material.

The presentation in Fig 1 (b) appear counter intuitive: why does the total REC appear to be higher for smaller features? Q , β_0 and β_k in the caption of Figure 1 are not defined at this point

Our reply:

Intuitively, the total REC quantifies the number of features—both large and small—that can be reliably estimated given a fixed sample size. As the sample size increases, larger features are resolved first, followed gradually by smaller ones. The total REC may appear higher when smaller features begin to be reliably estimated because it also includes the larger features, which require fewer samples to resolve. Once smaller features can be reliably estimated, the larger ones are already well resolved.

We have updated Fig. 1 to clarify that the Rayleigh Resolvable Features and Sub-Rayleigh Features are classified along the y-axis direction. The right part of the plot now correctly includes both types of features.

We have revised the description when referring to Fig. 1 of the main text to avoid this potential confusion: “Intuitively, the total REC quantifies the number of reliably estimated features—both large and small—and increases as smaller features become resolvable.”

We have moved the explanation of the learning formalism to combine with Eq. 1 in the main text. Now Q , β_0 and β_k in the caption of Figure 1 are introduced before the figure.

Page 3: the cluster of equations between 2-6 are not well-explained.

Our reply:

We have now added a separate section, “Preliminary”, to introduce the quantum learning formalism, with the aim of making the physical meaning of each symbol easier to follow.

Page 3, line 29 refers to this work as a 'letter'; same goes for page 3 of the supplemental material, line 12.

Our reply:

We now refer to this study as ‘this work’ throughout the manuscript.

I am unable to follow much beyond page 3: from page 3 onwards as the manuscript appear to be a discussion of dense symbols that are not very well-discussed.

Our reply:

We have significantly rewritten the entire manuscript to create a more coherent and structured presentation. For both the single and multiple compact source imaging cases, we now separate the discussion of eigenvalues, eigenvectors, and total REC to make the presentation more structured and easier to follow. Additionally, we include a new concrete simulation example to clarify the physical meaning of our results and to explicitly demonstrate the advantages of the quantum learning approach and our orthogonalized SPADE method. We also include, at the beginning of the Supplemental Material, a summary of the notations—organized by section of the discussion—to help readers follow the analysis.

Supplemental material:

$I(u)$ in Eq 1 is not defined.

Our reply:

We have added the definition of $I(u)$ below Eq. 1 of the Supplemental Material as: “Assume the normalized source intensity $I(u)$ is confined within the interval $[-L/2, L/2]$ ”

Eq: 12, a more intuitive/physical interpretation of the different beta needs to be given

Our reply:

We have added the physical interpretation of the β_k^2 under Eq. 15 of the Supplemental Material: “The REC of each eigentask is given by $C[f_k(\theta)] = 1/(1 + \beta_k^2/S)$. Intuitively, the total REC C_T accounts for all possible basis functions. In practice, however, most basis functions contribute negligibly to C_T , and it suffices to consider only the eigentasks f_k constructed from the vectors r_k that have significant contributions. Importantly, in the finite-sample regime, the relevance of each eigentask is well quantified by its associated eigenvalue β_k^2 .”

We thank the referee for the valuable comments, which have led to a substantial revision of the manuscript. All the points raised have been carefully considered. Our detailed responses are provided below, with the corresponding revisions highlighted in blue in the main text and Supplemental Material, and described in each reply.

Report of the Second Referee

Referee's comment:

The authors discuss an important application of quantum learning theory to the problem of superresolution imaging. By combining analytical approximations with numerical computations and simulations, they assess the impact of finiteness of the number of samples drawn from the probability distribution of specific POVMs to estimate the parameters of a one-dimensional (1D) optical source, specifically their spatial moments, from such data samples. Their choice of POVMs generalizes those proposed in two of the authors' earlier work (Ref. 5), but instead of Fisher information here they use the notion of resolvable expressive capacity (REC), introduced previously in Ref. 37, to quantify the number of degrees of freedom that can be accurately estimated with increasing sample size. They have employed the REC metric to contrast the generally inferior performance of direct imaging with superior superresolution protocols that employ wavefront projections, particularly in the sub-Rayleigh regime of source brightness features. They have introduced a new concept of orthogonalized SPADE basis that is optimized very generally, including the case of multiple compact sources that are in close proximity of one another regardless of each source's size and their mutual separation in relation to the Rayleigh resolution scale.

Here are some major observations that the authors must consider in improving the paper from its current form:

1. While the general results are presented with clarity, their over-reliance on and over-referencing of derivations presented in a large Supplemental document sometimes tends to obscure how these results connect with one another. I understand the need for brevity in the main paper, but I believe clarity can be improved in this respect with the inclusion of only a little extra text in places. An example of this kind of "disconnect" is between Eqs. (1) and (2)-(5), which are separated by a lot of text. I recommend that the authors bring forward the latter set of equations ahead of the paragraph entitled "Resolving two-point sources." In fact, a brief review of the REC concept covered in these equations with emphasis on the physical significance and origin of the various matrices in a separate section, before its applications to the different problem scenarios are discussed, will be much appreciated by the general reader.

Another instance of a lack of continuity is a discussion of Fig. 3 before the actual twin source structure to which the figure applies is first introduced in the following paragraph (which starts as "We perform a numerical..."). The order of these two discussions needs to be flipped.

Our reply:

We have now added a separate section, "Preliminary," to introduce the quantum learning formalism, with the goal of clarifying the physical meaning of the equations before delving into the detailed calculations in each case. The motivation and physical intuition behind applying this quantum learning approach are emphasized around Eqs. 1–5 as well as in the revised introduction, which we hope improves readability.

We have also moved the discussion of Fig. 3 and the corresponding numerical details directly after Eqs. 11 and 14 in the main text.

To improve the flow of the manuscript, we have reduced excessive referencing to the Supplemental Material and aimed to make the main text more self-contained.

Besides the points raised by the referee, we have also made the following revisions to improve the overall

clarity and flow of the paper. We added a concrete example in the new section “Demonstrative example” along with the newly added Sec. VIII in the Supplemental Material, to illustrate the advantages of the quantum learning approach and our orthogonalized SPADE method. We improved the clarity of individual sentences and provided more detailed explanations before introducing symbols and new concepts to ensure smooth transitions between sections. Additionally, the presentation of the calculations has been reorganized to separate different aspects of the analysis, making the discussion easier to follow, enhancing coherence, and reducing reliance on the Supplemental Material.

2. Most of the discussion of the paper is centered on scaling laws for the squared eigenvalues β_k^2 that govern the fidelity of estimation of the various parameter combinations. Apart from the case of a symmetric point-source pair, no specific, quantitative statements about the prefactors associated with these scaling laws are made, however. The numerical results that are presented in a number of figures certainly contain these prefactors numerically, but little attempt has been made to clarify their general form or dependences on the various source parameters. Their relative strengths affect the C_T curves, e.g., in making direct imaging perform better than SPADE in some ranges of the sample size, S , as seen in Fig. 4. This is a serious shortcoming of the presentation, but one which can be remedied by presenting some further discussion of these prefactors, even if it is only at the level of their numerically extracted values.

Our reply:

We have now added a new Sec. VII in the Supplemental Material to discuss the prefactors and briefly mention it in Page 4 of the main text: “The prefactors depend on the imaging strategy and the prior information ... of the Supplemental Material.” The prefactors of β_k^2 depend on both the prior information and the measurement strategy and are, in general, difficult to derive analytically. Interestingly, however, there exists a special case where the influence of the prior information and the measurement strategy becomes particularly transparent. This special case is characterized by two matrices: the matrix g in Eq. 26 of the Supplemental Material, which quantifies the prior information of the moments, and a newly defined matrix \mathbf{C} in Eq. 100, which primarily depends on the measurement strategy, such as direct imaging or SPADE methods. When g and \mathbf{C} commute, the product of their respective eigenvalues determines β_k^2 .

We demonstrate that, in some examples, this product provides a good approximation even when g and \mathbf{C} do not strictly commute. Although this approximation is introduced merely to build intuition about the behavior of the prefactors of β_k^2 and may not hold in general, it illustrates how the prefactor is influenced by both the measurement strategy and the prior information. In particular, the vectors C_i defined in Eq. 29 and Eq. 48 of the Supplemental Material, which constitute \mathbf{C} , are the coefficients of each order term in the series expansion of the probability distribution in powers of α . Intuitively, larger eigenvalues of \mathbf{C} indicate that the C_i are more independent, meaning that the probability distributions exhibit more distinct behavior in each coefficient. Consequently, the measurement outcomes may carry more information, as reflected by the larger eigenvalues of \mathbf{C} . And for a given measurement strategy, the prefactors are primarily determined by the fluctuations of the eigenvalues of g arising from different prior information.

This analysis thus provides intuition about how the prefactors of β_k^2 are shaped by the interplay between the measurement strategy and the prior information.

3. It is also important to stress in the paper that the total REC is independent of the choice of the complete basis of the f -functions, and only depends on the physical system and the POVM used to acquire the data. In this connection, it may also be worth pointing out if attempts have been made, perhaps by

others, to maximize the total REC over all possible POVMs to give it a truly invariant property of a given parametrized quantum state.

Our reply:

We think this is an interesting idea. Indeed, the total REC is independent of the choice of the complete basis of the f -function and depends only on the physical system and the POVM used in the detection. However, the total REC does depend on the choice of POVM. We reviewed recent papers since the proposal of the REC formalism but, unfortunately, found no discussion of the total REC optimized over all possible POVMs.

The total REC is given by $C_T = \text{Tr}((G+V/S)^{-1}G)$, where $D_{ii} = (M_i \rho^{(1)})$ and $G_{ij} = ((M_i \otimes M_j) \rho^{(2)})$, $\rho^{(t)} = \int \hat{\rho}(\theta)^{\otimes t} p(\theta) d\theta$. We aim to optimize over all possible POVMs M_i to maximize C_T for each possible S . This is a highly nontrivial optimization problem since it involves the matrix inverse and a bilinear function of M_i . This difficulty might be expected, as the optimal measurements for different features may require mutually incompatible measurements. In the case of quantum Fisher information, which optimizes over all possible POVMs, the quantum Fisher information matrices can be computed, but their bounds are often unattainable due to incompatibility when estimating multiple parameters.

In the case of the REC formalism, the focus is not on estimating parameters but rather on identifying the features or eigentasks that can be well approximated with a given measurement. However, similar difficulties might arise if we attempt to consider the precision of all measurable features. A reasonable approach might be to consider optimizing the REC in certain special cases—for example, when all the features can be estimated using compatible measurements. However, this would require considerably more work; for example, it would be necessary to clarify the conditions under which all the features can be estimated using compatible measurements in this setting. Therefore, while this direction seems interesting, we currently do not have a good idea of how to find a closed-form expression for this optimization. It might be better to leave this as future work.

We now mention this above Eq. 2 of the main text: “The total REC $C_T := \sum_k C[g_k]$ fulfills this purpose, where $\{g_k\}_k$ can be any complete orthonormal basis of functions in the Hilbert space equipped with inner product $\mathbb{E}_\theta[g_k(\theta)g_\ell(\theta)]$.”

We try to add some sentences in the “Conclusion” section of the main text to motivate future works in this direction: “Note that the total REC depends on the POVM used in the detection. It may be interesting to explore whether a closed-form expression of the total REC optimized over all possible POVMs can be found, at least in some special cases—for example, when the measurable features require compatible measurements.”

4. I also do not see how the authors can assert that for SPADE-based superresolution, the n th estimated function, $\sum r_{nm} P_m$, only involves the $2n$ th and $(2n+1)$ th moments in the limit $\alpha \rightarrow 0$. For this to be true, the matrix of eigenvectors r_{nm} has to be nearly diagonal, up to permutations, in the limit $\alpha \rightarrow 0$. Otherwise, each estimated function would be a linear combination of many moments that is dominated by the lower-order moments in that limit. This point needs clarification in the second half of the paragraph containing Eq. (8), since the parallel with direct imaging discussed in the previous paragraph is unclear.

Our reply:

The eigenvectors are obtained using perturbation theory. Intuitively, this follows from the fact that G

has the following structure

$$G = \sum_{n=0}^{\infty} G^{(n)} = \sum_{n=0}^{\infty} \alpha^n \sum_{i+j=n} g_{ij} C_i C_j^T, \quad C_i = \Pi_{\lfloor i/2 \rfloor} C_i^0, \quad (1)$$

$$C_i^0 = [c_i, c_{0i}^+, c_{0i}^-, c_{1i}^+, c_{1i}^-, \dots]^T, \quad \Pi_m = \begin{bmatrix} I_{2m+3} & 0 \\ 0 & 0 \end{bmatrix}.$$

where, if we consider terms up to order $O(\alpha^n)$, the remaining elements of G render it a rank-deficient (not full-rank) matrix. For example, if we retain only terms up to order α^2 ,

$$G = g_{00} C_0 C_0^T + \alpha(g_{01} C_0 C_1^T + g_{10} C_1 C_0^T) + \alpha^2(g_{20} C_2 C_0^T + g_{11} C_1 C_1^T + g_{02} C_0 C_2^T) + o(\alpha^2) \quad (2)$$

Such a G , truncated at order $\Omega(\alpha^n)$, can obviously yield the first few eigenvalues, since the remaining parts of G are much smaller and therefore contribute only to higher-order eigenvalues. Furthermore, observe the structure of $C_i = \Pi_{\lfloor i/2 \rfloor} C_i^0$, which has vanishing elements at certain positions, which is a key difference from the direct imaging case. Recall that the eigenvectors satisfy $Gr_k = \lambda_k Dr_k$ in Eq. 47 of the Supplemental Material, where D is diagonal, and the structure of r_k is primarily determined by G . Because G has a rank-deficient structure when truncated to a fixed order in α , the eigenvectors are directly related to the vector set obtained from Gram–Schmidt orthogonalization of the set $\{C_0, C_1, C_2, \dots\}$. And because of the structure of the vectors C_i , which have a triangular form when arranged as the columns of a matrix, this in turn induces the triangular structure of r_{nm} . Consequently, the higher-order eigentasks are dominated by higher-order moments in the case of imaging a single compact source using SPADE method. This intuition is rigorously validated using perturbation theory in Sec. III B of the Supplemental Material.

We also numerically confirm this eigenvector in Sec. III D of the Supplemental Material, as shown in the updated Fig. 4 of the Supplemental Material. In Fig. 4(b2)(b3) of the Supplemental Material, we visualize the distribution of the eigenbasis for the SPADE method. It is evident from Fig. 4(b2)(b3) of the Supplemental Material that the eigenvector is dominated by only a few components in the limit $\alpha \rightarrow 0$, as we have claimed. This observation supports the analytical analysis based on perturbation theory presented in Sec. III B of the Supplemental Material. In Fig. 4(b1) of the Supplemental Material, we also plot the deviation from the actual eigenbasis when retaining only the two leading-order terms. We observe that this deviation vanishes as $\alpha \rightarrow 0$.

We provide some explanations for this structure in Page 5 of the main text: “The \$n\$ th eigentask corresponds to \$\sum_m r_{nm} P_m\$ as a function of \$\vec{x}\$, ... The first two leading-order terms of both the \$2k\$ -th and \$(2k + 1)\$ -th eigentasks have coefficients \$x_{2k}\$ and \$x_{2k+1}\$.”

We included the intuitive interpretation in Sec. III B of the Supplemental Material, before the full proof based on perturbation theory: “In the following, we analyze the spectrum of the SPADE method using perturbation theory. ... This intuition is rigorously validated by the perturbative analysis presented below.”

We numerically confirm the eigenvectors in the updated Fig. 4 of the Supplemental Material and provide additional explanation in Sec. III D of the Supplemental Material: “We also plot the actual eigenbasis \$y_k\$ and \$r_k\$ in Fig. 4(a2)(a3) for the direct imaging method and in Fig. 4(b2)(b3) for the SPADE method.... Using \$y_k\$ makes it easier to see which components become dominant.”

5. The presence of a slowly increasing plateau in total REC (Fig. 2(c)) as $S \rightarrow \infty$ for compact sources that are large compared to the PSF width needs further examination, perhaps by means of an asymptotic approximation of the total REC in this limit. I believe that the plateau would never stop rising with increasing S , but its increase would continue to maintain a logarithmic type of dependence on S with a

*relatively small coefficient. Indeed, in the limit of infinitely many photons even direct imaging in the absence of noise should be able to superresolve any compact source to an arbitrary degree of fineness - and thus attain arbitrarily large values of C_T - regardless of the PSF width, as a number of investigations over the last fifty years have shown (see, e.g., for superresolving a point-source pair, Lucy, L., “Statistical Limits to Super Resolution,” *Astron. Astrophys.*, vol.261, 706-710 (1992)).*

Our reply:

We agree with the referee’s comments. If we make $S \rightarrow \infty$, the total REC C_T will continue increasing slowly and eventually reach infinity. This means that as long as we have infinitely many photons, even direct imaging can resolve the images with an arbitrary degree of fineness. If we treat C_T as a function of $\log S$, where S is the sample number, we add the new Fig. 5(c) in the Supplemental Material to show that the slope of this function $\partial C_T / \partial \log S$ reaches a peak at some sample number and gradually decreases. In this sense, the referee is right that C_T is increasing in a logarithmic-type dependence on S with a relatively small coefficient when S is sufficiently large.

Since $\log S$ is used as the x -axis in this slope plot, when $\log S$ becomes sufficiently large, advancing by a fixed interval along the $\log S$ axis requires an exponentially larger number of additional samples. Meanwhile, C_T exhibits only a modest increase over the corresponding interval of $\log S$. In this sense, the curve approaches a slowly rising plateau.

We emphasize that the relation between the total REC C_T and L/σ is intended only as a heuristic example to illustrate the application of REC, rather than a formal or rigorous result. Since we now provide a new concrete simulation example to explain the physical meaning of the quantum learning formalism in the newly added section “Demonstrative example,” we have moved the original discussion of Fig. 2(c) to Sec. IV of the Supplemental Material.

We have revised the statement where we refer to Fig. 5(a) of the Supplemental Material (original Fig. 2(c) of the main text) to clarify that the relationship between C_T and L/σ is approximate and not exact: “The total REC represents the number of degrees of freedom we can extract from the measurement, and since the image is blurred, it is roughly on the order of L/σ ; however, this relation is not exact, as it arises from heuristic reasoning rather than formal analysis.”

We have added Fig. 5(c) in the Supplemental Material and the following explanation to explain such effect in Sec. IV of the Supplemental Material: “We plot the total REC as a function of sample number S for different values of L ... C_T increases only modestly over the same interval of $\log S$. Therefore, the curve approaches a slowly rising plateau. ”

A number of minor errors and comments are listed below.

1. Page 1, line 72, over what space is the set of functions $\{f_l\}_l$ orthonormal and complete? I imagine it is the space over which the system parameters $\langle \theta_b \rangle$ live, but this should be simply stated.

Our reply:

$\{f_l(\boldsymbol{\theta})\}_l$ is the set of complete and orthonormal basis functions in the Hilbert space of all functions of input parameters $f(\boldsymbol{\theta})$ equipped with inner product $\mathbb{E}_{\boldsymbol{\theta}}[f_k(\boldsymbol{\theta})f_l(\boldsymbol{\theta})]$. Note that this set of basis functions has infinitely many elements, as the space of functions of $\boldsymbol{\theta}$ is infinite-dimensional. However, the calculation of $C_T = \sum_{l=0}^{\infty} C[f_l]$ can be reduced to just the eigentasks $C_T = \sum_{k=0}^{K-1} C[f_k]$, defined based on $Vr_k = \beta_k^2 Gr_k$, $f_k(\boldsymbol{\theta}) = \sum_m r_{km} \eta_m(\boldsymbol{\theta})$, where K is the dimension of matrices G and V . Intuitively, C_T includes the REC

for all possible basis functions. In practice, however, most basis functions do not contribute to C_T , and we only need to include the eigentasks f_k , which actually contribute. This was proven in Hu et al., Physical Review X 13, 041020 (2023), as cited above Eq. (2) in the main texts.

We added explanation above Eq. 2 of the main text: “The total REC $C_T := \sum_k C[g_k]$ fulfills this purpose, where $\{g_k\}_k$ can be any complete orthonormal basis of functions in the Hilbert space equipped with inner product $\mathbb{E}_\theta[g_k(\theta)g_\ell(\theta)]$.”

We try to mention this above Eq. 5 of the main text: “The eigenbasis r_k correspond to a minimal set of eigentasks $f_k(\theta) = \sum_m r_{km}P_m(\theta)$ that saturate the available REC of the system in the space of all functions of input parameters θ .”

We also provided more details about this point above Eq. 13 of the Supplemental Material: “The total REC, $C_T = \sum_{i=0}^{\infty} C[f_i]$, ... eigentasks defined based on the following eigenvalue problem.” and after Eq. 15 of the Supplemental Material: “The REC of each eigentask is given by $C[f_k(\theta)] = 1/(1 + \beta_k^2/S)$... the relevance of each eigentask is well quantified by its associated eigenvalue β_k^2 .”

2. Line 49, page 2, the Gaussian function, $\phi_0(u)$, has the power of 2 missing from u inside the exponent.

Our reply:

We have corrected the typo.

3. Line 92, page 3, the relation $\beta_k^2 = \Theta(\alpha^{2\lceil k/2 \rceil})$ has a negative sign missing in the exponent.

Our reply:

We have corrected the typo, but the equation has been moved to Sec. III C of the Supplemental Material as a result of the rewriting of the analysis in the main text.

4. Line 26, page 5, a similar relation has not just the minus sign missing in the exponent but, I believe, the $\lceil k/Q \rceil$ should be replaced by $\lfloor k/Q \rfloor$ so this relation becomes consistent with Eq. (11).

Our reply:

The referee is correct about this relation, and we have now corrected the expression accordingly.

5. Line 56, page 5, the expression for β_k^2 is incorrect. I believe it should be $\beta_k^2 = \Theta(\alpha^{-2\lceil \lfloor k/Q \rfloor / 2 \rceil})$ to be consistent with Eq. (14).

Our reply:

The referee is correct about this relation, and we have now corrected the expression accordingly.

6. Two instances of power 2 are missing in the last two expressions giving approximations for S in the caption of Fig. 1.

Our reply:

We have corrected the typo.

7. In Fig. 4, the dotted and dashed curves are hard to see clearly in some of the plots. I suggest making them thicker and/or reducing the thickness of the solid curves.

Our reply:

We have updated the Fig. 4 of the main text to make the curves thicker.

8. Line 33, page 7, replace "a issue that are" by "an issue that is." Similar grammatical errors occur elsewhere too. Authors, please look for them.

Our reply:

We have corrected the errors in this section and carefully reviewed the entire paper to check for additional grammatical issues.

9. Line 64, page 3, change "providing" to "provides."

Our reply:

This part of the discussion has been rewritten to improve clarity.

The paper discusses an important topic from the new perspective of sample-based capacity of quantum measurements on a system state to extract certain linear combinations of the state parameters accurately. If the authors can sufficiently address all of the points I have raised here regarding the current version of the manuscript, then I can recommend its publication in Nature Communications.

Our reply:

We have carefully considered all of the referee's comments and revised the manuscript accordingly. Given the two main contributions of our work—introducing the quantum learning approach to quantum imaging and proposing the orthogonalized SPADE method to address more general and complex practical imaging tasks—we hope the referee finds the revised manuscript substantially improved and suitable for publication in Nature Communications.

We thank the referee for the valuable comments, which have led to a substantial revision of the manuscript. All the points raised have been carefully considered. Our detailed responses are provided below, with the corresponding revisions highlighted in blue in the main text and Supplemental Material, and described in each reply.

Report of the Third Referee

Referee's comment:

The authors present a learning theory approach to the expanding field of quantum-inspired superresolution. In particular, they consider the resolvable expressive capacity (REC) as an image quality quantifier. The total REC represents the total number of linearly independent functions that can be expressed as a linear combination of the measurement results. This quantity presents some interesting properties that allow to identify how the ability to learn sub-diffraction features of an object depend on the sample size. In particular, one can observe jumps in the dependence of the total REC with the sample size, and the amplitude of these jump depend on the kind of measurement that is performed.

I find interesting how this result agrees with the prediction of quantum-inspired superresolution theory that spatial mode demultiplexing (SPADE) techniques can overcome the resolution of direct imaging. I also find theoretically interesting how one can use the total REC to gain an intuition on how much one can divide an extended source in terms of subdiffraction objects to be analyzed by the appropriate orthogonalized SPADE techniques. However, I have a hard time to understand the operational meaning of these results. In other words, how can one relate the REC with a procedure to reconstruct an image of the object with super-resolution features? Without an answer to this question, I cannot recommend publication in Nature Communications.

I also think that the bibliography is a bit outdated, and there several technical details that should be clarified before resubmitting to Nature Communication or elsewhere. I will detail these, point by point in the following.

Our reply:

We begin with a general discussion comparing our approach with the Fisher information matrix (FIM) method and clarifying the operational meaning of the quantum learning approach adopted in this work. This is presented in the newly added section “Demonstrative example” in the main text, with further details provided in Sec. VIII of the Supplemental Material. We emphasize that our work introduces the quantum learning framework to quantum imaging as a systematic approach to addressing imaging tasks, rather than simply using the total REC as a measure of estimation precision. The quantum learning approach adopted in our work, encompassing both the training process and the application of the trained model for inference, offers a systematic methodology and clear theoretical guidance on how complex imaging tasks should be addressed with rigorous performance guarantees and quantitative benchmarks. Machine learning has achieved tremendous success in imaging applications, enabling the handling of complex and practical tasks—even though, in principle, all imaging problems can be modeled as parameter estimation based on Fisher information. This is because that machine learning provides effective tools for addressing the complexity and ambiguity often present in real-world imaging scenarios. We believe that quantum imaging can similarly benefit from learning-based approaches. The quantum learning approach adopted in our work inherits key strengths from machine learning and offers additional advantages beyond the two points mentioned by the referee, as detailed below.

We emphasize key limitations of the FIM approach in imaging: when the image task is framed as parameter estimation, there are at least two sources of ambiguity. First, it is often unclear which set of

parameters is most appropriate. For a practical imaging task, should we choose image moments, Fourier components of the source, or the intensity of each pixel on the source? Each of these forms a complete set of parameters containing the full information about the image. For instance, in face identification tasks, determining the optimal set of parameters to represent a person’s identity is itself a nontrivial problem. Different choices of parameters lead to different values of the Fisher information matrices. In contrast, the REC formalism, particularly the eigentasks and the total REC, is invariant under different parameterizations. Second, after selecting a parameter set, finite sample sizes mean that some parameters can be estimated accurately while many others remain poorly estimated and dominated by noise. To make practical use of the data, it is essential to identify which parameters are well estimated so that only these can be retained for downstream analysis. The REC formalism provides precisely this by identifying the sample threshold for each eigentask. One major advantage of machine learning is its ability to accomplish complex practical tasks. This highlights why the quantum learning approach offers a more general framework than the FIM approach. In fact, a notable limitation of prior superresolution studies based on FIM is their reliance on a fixed set of parameters, often without a strong justification for their selection. While the FIM provides a rigorous and informative bound for the precision of estimating well-defined parameters, its applicability is fundamentally restricted to parameter estimation problems.

In the broader context of quantum learning, the REC formalism is developed within a paradigm known as physical neural networks (PNNs), where an analog physical system is used to compute a function of input parameters. Instead of explicitly designing a circuit to perform a desired operation, PNNs employ a fixed physical dynamical system with inherently complex dynamics. Inputs are encoded into this system, and its evolution maps them to measured features in a high-dimensional space. Because the dynamics are rich and complex, the resulting feature set can approximate a wide range of functions through linear combinations of these features. Consequently, only the output weights need to be trained—typically using simple methods such as linear regression for regression tasks or logistic regression for classification tasks—while the internal structure of the physical system remains fixed. This approach is particularly suitable for implementation on NISQ devices because it avoids the need for deep circuits or extensive parameter optimization, which are challenging under current hardware limitations. By leveraging the natural dynamics of the physical system, it reduces training complexity while still providing rich feature representations for learning tasks. We have incorporated a more detailed introduction to PNNs in Sec. I B of the Supplemental Material, as well as in the revised introduction of the main text.

The relationship between a specific physical system and the classes of functions it can accurately express is a fundamental question in PNNs. The REC formalism was developed to identify such resolvable functions with quantified precision in the presence of finite sample noise. In this framework, the eigentasks constitute a set of functions that span the space of approximable functions. They thus indicate which set of features is most suitable for a given imaging task, as any other measurable function can be expressed as a linear combination of these eigentasks. This directly addresses the first ambiguity of the FIM approach. Importantly, the REC formalism is invariant under reparameterization, as explicitly proven in Proposition 1 of the Supplemental Material. The features relevant to the imaging task are identified directly from the measurement, the physical system, and the problem’s structure, without relying on an artificial choice of parameterization. Furthermore, the eigenvalues β_k^2 specify the sample size needed to reliably estimate each eigentask, and the total REC C_T determines how many eigentasks can be retained for downstream analysis. By providing a principled criterion for determining how many eigentasks to retain in downstream analysis, the REC formalism also resolves the second ambiguity of the FIM approach. Importantly, selecting low-noise eigentasks can substantially improve the performance of machine learning tasks such as classification. This advantage, highlighted in the original paper [PRX 13, 041020 (2023)], is also very relevant for imaging tasks.

We emphasize that the strength of the quantum learning approach lies in its ability to treat complex

practical imaging tasks as learning problems within the PNNs framework, where a systematic methodology can be applied. The first step is to identify eigentasks that are well estimated from the measurement, the physical system, and the structure of the problem—without relying on an artificial choice of parameterization. The PNN model then addresses practical tasks by training its output weights on the training data and subsequently applying the trained model for inference on the experimental data. We adopt this same strategy to tackle complex imaging problems, thereby establishing a systematic framework with clear theoretical guidance, rigorous performance guarantees, and quantitative benchmarks. These capabilities to address complex practical imaging tasks motivate adopting the REC formalism for imaging, rather than relying solely on traditional Fisher-information-based methods.

To demonstrate the advantages of the quantum learning approach in addressing complex practical problems, clarify its operational meaning, and highlight the benefits of our orthogonalized SPADE method, we introduce a new section, “Demonstrative example”, in the main text, with further details provided in Sec. VIII of the Supplemental Material.

We introduce a distinguishing task in which a figure is randomly selected from one of two distinct sets, each generated by a different procedure and consisting of randomly varying figures unique to that procedure. The goal is to determine whether a given figure was produced by the first or second procedure—an abstractly simplified version of QR code identification. As explained above, we first identify the eigentasks determined by the imaging system and the problem structure, and then train the output weights using the training data. The success probability is subsequently evaluated on a separate test dataset with the trained model. This provides a complete example of how the REC formalism can be applied to address a complex imaging task. Although this example focuses on distinguishing tasks, the same method possesses the generality to address a wide range of learning problems, including regression and clustering, making it broadly applicable to practical imaging scenarios.

As shown in Fig. 20 and Fig. 22 of the Supplemental Material, the way these random figures are generated can strongly influence the structure of the eigentasks identified by the quantum learning formalism, which are generally not simple or intuitive. These eigentasks, along with their corresponding eigenvalues β_k^2 , play a central role in the distinguishing task. Any image feature—expressed as a function of input parameters and well approximated by a linear combination of measured features—must lie within the subspace spanned by the reliably estimated eigentasks, given a finite sample size S . Thus, the eigentasks and their spectrum β_k^2 have clear operational meaning: β_k^2 sets a threshold for sample size, such that when $S \gg \beta_k^2$, the k th eigentask can be reliably estimated.

For this distinguishing task, we use logistic regression to train the output weights on the training data, yielding a function of the eigentasks that approximates the classification rule. The trained model is then applied to the test dataset to assign each image to one of two categories. It is crucial to include only those eigentasks that can be reliably estimated given the available sample size. As shown in Fig. 6 of the main text, increasing the number of included eigentasks initially improves the success probability, which then begins to decline. This behavior is intuitive: adding more eigentasks at first captures additional information about the intensity distribution, enhancing classification performance. However, beyond a certain point, higher-order eigentasks become poorly estimated due to the limited sample size S . Naively including these noisy eigentasks degrades the performance of the classifier and ultimately reduces the success probability.

This simulation highlights the operational meaning of the total REC C_T . First, C_T estimates the number of eigentasks that can be reliably included in the training of output weights, such as logistic regression in our example. Since β_k^2 sets the sample threshold for estimating the k th eigentask and $C_T = \sum_k 1/(1 + \beta_k^2/S)$, with each term approaching 1 when $S \gg \beta_k^2$, C_T effectively counts the number of reliably estimated eigentasks. Second, a larger C_T allows more well-estimated eigentasks to be used, capturing more information and improving classification performance; hence, a larger C_T reflects better imaging performance. This second

operational meaning of C_T motivates the use of C_T as a performance metric for imaging, but the quantum learning approach provides a more systematic and powerful framework that goes beyond simply evaluating C_T .

In contrast, applying the FIM approach to this classification problem presents several challenges. In our distinguishing task, the goal is to classify figures generated by two distinct procedures—rather than to estimate any specific parameters. This raises an immediate difficulty for the FIM approach: it is unclear which parametrization is most appropriate for capturing the characteristics of such a complex imaging task. Moments, Fourier components, and the source intensity at each pixel can all serve as complete sets of parameters, but it remains unclear how to decide which of these parametrizations best indicates the performance of the distinguishing task considered here. Furthermore, even if we adopt a specific parameterization, it remains unclear how many parameters—such as up to which order of moments—should be included in the downstream analysis. Note that including poorly estimated parameters can degrade the performance of downstream analysis, such as the logistic regression in this example. There is no obvious criterion for selecting a parameter set that meaningfully reflects imaging performance in this context. We believe this reflects a common situation in practice, where imaging is motivated by a practical objective, but there is no clearly defined set of parameters to be estimated. Relying solely on the FIM and traditional parameter estimation thus requires careful, case-by-case choices, lacking both conceptual simplicity and a unified theoretical framework.

So, we believe the quantum learning approach offers a systematic and powerful framework for addressing imaging problems, with a clear operational meaning that both quantifies imaging performance and guides downstream analysis in complex, practical scenarios. It identifies which features of the image can be reliably estimated, even in the presence of complex physical models and finite sample size. This framework is expected to demonstrate significant advantages over the FIM approach in many practically relevant imaging tasks. This is particularly evident in imaging, where the source’s intensity distribution often has infinitely many degrees of freedom, making the identification of relevant features both challenging and crucial.

We would also like to highlight to the referee that another important contribution of our work is the introduction of the orthogonalized SPADE method. This represents a nontrivial generalization of the original SPADE approach. Previous superresolution studies typically assume the entire source is much smaller than the Rayleigh limit—a single compact source—whereas practical imaging often involves sources whose overall size exceeds this limit, with fine features of interest smaller than the Rayleigh limit. To bridge this gap, we consider multiple compact sources, motivated by the idea that partitioning a large source into smaller pieces and applying SPADE method locally might improve performance. This is equivalent to imaging multiple compact sources. However, when these sources are spaced closely, they introduce significant measurement noise, and the naive implementation of SPADE used in earlier work, referred to as the separate SPADE method, fails to outperform direct imaging. Our orthogonalized SPADE method partially addresses this issue by allowing compact sources to be placed closer—comparable to the PSF width in our tests—whereas the naive separate SPADE method requires much larger separations.

Therefore, we claim that imaging multiple compact sources constitutes a nontrivial generalization of the earlier SPADE method, representing a step toward making the superresolution approach more practical. At the very least, our measurement construction enables us to handle certain general images whose overall size exceeds the Rayleigh limit but whose structures remain somewhat clustered. In this sense, we aim to provide a generalization of the superresolution approach that is more practically useful. We hope that this discussion lays the groundwork for imaging general sources whose overall size exceeds the Rayleigh limit, while still resolving fine details of the source using superresolution techniques.

The bibliography is updated following the comments. Below, we also address the two points raised by

the referees.

Referee’s comment:

The authors identify to main advantages of the REC over the quantum Fisher information matrix (QFIM): 1) the REC is a scalar quantity with a clear interpretation, while to reduce the QFIM to a scalar one needs to introduce a weight matrix 2) the QFIM is relevant in the asymptotic regime, while the REC is valid also in the finite sample regime.

Regarding 1): the QFIM gives a clear and complete (up to incompatibility issues) picture of a multi-parameter estimation problem, providing a lower bound on the parameters’ covariance matrix. Most importantly, it gives a bound on a precise image reconstruction method which consists in estimating a given number of parameters fully defining the observed object. If one needs a scalar metric one can use a weight matrix, but in principle one has the full information in the QFIM. On the other hand, which is the learning task that allows to reconstruct the object from the measurement data? What does the REC tells us about how well can we do this? Can the authors reply to these questions? If not, I think that the clear operational meaning of the QFIM makes it a much better quantifier for the performances of an imaging task.

Our reply:

Some advantages of our approach over the Fisher information matrix (FIM) approach have already been clarified above in the general discussion, but we summarize the relevant points below in response to this question.

The FIM approach faces at least two key ambiguities: first, it is unclear which parametrization best captures a complex practical imaging task; second, even after selecting a parametrization, the combination of finite sample sizes and a large number of parameters makes it difficult to identify which subset of parameters is well estimated and should be retained for downstream analysis. These ambiguities are exemplified by the distinguishing tasks presented in our simulation in the newly added section “Demonstrative example” in the main text, with further details provided in Sec. VIII of the Supplemental Material. Current studies of imaging based on the FIM approach typically assume a fixed set of parameters without providing a compelling justification for their selection. This limitation is significant and, in our view, acceptable only at the early stages of developing superresolution techniques. Our work seeks to advance the discussion by adopting a more general perspective.

We also emphasize that the spectrum β_k^2 and the eigentasks together encapsulate the full information about imaging performance. While a scalar metric is useful for summarizing overall performance, the spectrum and eigentasks offer deeper insights and guide the practical use of the full information for complex tasks. Specifically, the eigentasks reveal which features can be reliably estimated given a finite sample size, with each β_k^2 indicating a threshold: when the sample number S satisfies $S \gg \beta_k^2$, the k -th eigentask can be accurately estimated.

This simulation illustrates the operational meaning of the total REC C_T . First, C_T estimates how many eigentasks can be reliably included in downstream tasks such as logistic regression. For effective analysis, one must include all well-estimated eigentasks—omitting them underutilizes the available information, while including poorly estimated ones introduces noise and degrades performance. Fig. 6 of the main text clearly illustrates this behavior. Second, a larger C_T allows more reliable eigentasks to be used, enhancing classification and indicating better imaging performance. The second operational meaning supports using C_T as a performance metric, but we would like to emphasize that the quantum learning approach offers

a broader and more powerful framework that goes beyond a single scalar measure. To tackle a complex practical imaging task, we first train a model on the training data and then apply it to the experimental data for inference. Our work demonstrates how such a learning-based framework can be effectively applied to complex imaging tasks, while C_T provides a useful additional insight into imaging performance within this broader approach.

Referee’s comment:

Regarding 2) it is true that the QFIM has a clear interpretation only in the asymptotic regime. However, the authors results show that for the total REC to be non-zero one needs thousands when not millions of samples, so is the asymptotic value of the QFIM a practical limitation in this contest? And if one was really interested in meaningful finite-sample results, why on should prefer a learning theory approach instead of a Bayesian estimation theory approach?

Our reply:

Some advantages of our approach over the Fisher information matrix (FIM) approach in the finite-sample regime have already been highlighted in the general discussion above. Here, we further summarize the key points and provide additional details in response to this question. As emphasized earlier, a fundamental limitation of the FIM approach lies in its reliance on formulating the practical problem explicitly as a parameter estimation task. From the outset, it can be challenging to determine which set of parameters should be chosen to adequately represent the practical application at hand.

Beyond this issue, we believe that the finite-sample problem is far from trivial. As shown in Fig. 6 of the main text, in practice, identifying which measured features can be trusted and used in downstream analysis is crucial, and a proper selection of well-estimated features is key to achieving optimal performance. The performance can in fact be quite sensitive to this selection; including too many or too few features can quickly degrade the performance. There is no guarantee that thousands or even millions of samples are sufficient to reach the asymptotic regime for estimating certain parameters. Even with millions of samples, as demonstrated in our simulation, it remains essential to carefully select which features to include in the downstream analysis. Some features are intrinsically difficult to estimate and require an enormous amount of data to build reliable statistics, and there is no guarantee that this can be achieved in practice.

What makes the situation even more challenging is that imaging inherently involves a large—and often infinite—number of degrees of freedom. In complex practical applications, many of these parameters may need to be examined to determine whether they have reached the precision limit suggested by the FIM, which is a nontrivial task. In contrast, the quantum learning approach offers a simple and systematic way to identify when each feature is well estimated.

Bayesian estimation theory is certainly a viable approach to address precision in the finite-sample regime. However, it faces challenges similar to those of the FIM approach, particularly for complex practical imaging tasks, where there is comparable ambiguity about which parametrization best captures the task to be addressed. In such cases, one must formulate the problem carefully within the Bayesian estimation framework on a case-by-case basis. Moreover, imaging problems intrinsically involve a large—often infinite—number of degrees of freedom, which makes the analysis particularly cumbersome. So, this formulation process often lacks simplicity and a general theoretical foundation for how to handle diverse and complex practical tasks. In contrast, the quantum learning approach offers a key advantage in its generality: it provides a systematic framework for addressing practical imaging problems. Its value lies in the ability to identify the eigentasks based on the prior information and the physical imaging system, and to quantify the precision with which each eigentask can be estimated with finite samples, in a manner that is independent

of parameterization. The learning approach then provides guidance for solving the problem by training a model using the eigentasks and applying the trained model to address the practical task.

Referee’s comment:

The bibliography is incomplete and outdated. In particular, the most recent experiments using multimode Hermite-Gaussian mode sorters are missing: Rouviere et. al. Optica 11 166 (2024) where the record sub-Reyleigh sensitivity in separation estimation has been reached, Tan et al. Optica 10 1189 (2023), where the moments estimation approach considered by the authors was implemented, Santamaria et al. Optica Quantum 2 46 (2024) where sources of different intensity have been considered.

Other papers that might be interesting for the authors are: Sorelli et al. Phys. Rev. Lett. 127 123604 (2021) where it was shown that estimators only based on linear combinations of the first moments of the measurement results can saturate the ultimate quantum limit. This seems to be similar to the authors observation that linear combinations of the measurement probability can be used to reconstruct moments of the object.

The authors distinguish between sub-Rayleigh features and features larger than the Rayleigh limit and discuss how different measurements are effective in the different regimes. They might be interested in the results of Grace et al. JOSA A 37 1288 (2020) where it was investigated how to switch between direct imaging and SPADE for the simultaneous estimation of separation and centroid.

Our reply:

Rouviere et. al. Optica 11 166 (2024), Tan et al. Optica 10 1189 (2023), Santamaria et al. Optica Quantum 2 46 (2024) report notable experimental progress on this problem. We have cited these works in the introduction, where we discuss the experimental advances in superresolution: “These theories have been experimentally demonstrated ... for estimating the moments of the sources [21].”

Sorelli et al. Phys. Rev. Lett. 127 123604 (2021) propose a useful estimator for resolving the separation between two point sources in the Rayleigh limit. However, they consider a very simple imaging scenario—two point sources—whereas we address the imaging of multiple general compact sources that can exhibit much more complex structures. Moreover, we introduce a quantum learning approach to this superresolution problem. Their work remains within the conventional framework of parameter estimation based on the Fisher information, focusing on the simplest superresolution task: estimating the separation between two point sources. Crucially, the estimation target in their work is a well-defined parameter, namely the distance between two point sources. In contrast, as discussed in the newly added section ‘Demonstrative Example’ of the main text, with further details in Sec. H of the Supplemental Material and in the above response, the quantum learning approach provides a systematic framework for imaging tasks where the objective extends beyond estimating a specific parameter. Note that our work is not solely aimed at estimating the moments; the moments are just one example of possible input parameters. Our goal is to determine which features can be reliably estimated, given the prior knowledge of the system and the measurement model in the finite-sample regime. In fact, the eigentasks identified for imaging multiple compact sources are quite general and are strongly influenced by the specifics of the actual problem. Our approach can therefore address complex practical problems, such as face identification, by guiding the choice of downstream data analysis strategies, making it more powerful and broadly applicable to real-world scenarios than the work of Sorelli et al.

Grace et al. JOSA A 37 1288 (2020) focuses on estimating the separation and centroid of two point sources of equal strength. In their work, direct imaging is used to estimate the centroid of the two sources,

while the separation is more accurately estimated using the SPADE method. Their discussion, however, is also limited to a simple parameter estimation problem. In contrast, our approach adopts a quantum learning framework to tackle the imaging problem and considers a general scenario, where the source can be more complex than just two point sources.

We have also cited the works of Sorelli et al. and Grace et al. in the introduction of the main text: “These extensions include more careful treatment of the measurement and data analysis for imaging two point sources [3,4]...”, as well as in a new paragraph in Sec. I A of the Supplemental Material, where we discuss their relation to our work: “We briefly summarize related theoretical works for comparison with our results. Sorelli et al. ... where the source can have structures far more complex than just two point sources. ”

Referee’s comment:

Another point that the authors should clarify is the role of the prior probability in the calculation of the REC. The only point, where the authors explicitly state their prior is in the two sources problem. In this context, they assume that the width of the prior is much smaller than the width of the point spread function of the imaging system. I find this assumption problematic, since it seems to suggest that one needs prior subdiffraction information to achieve subdiffraction resolution. Could the authors clarify if this the case and which assumptions on the prior probability are made in the more general scenarios considered in the paper?

Our reply:

We choose such a prior mainly because the advantage of a superresolution method over direct imaging becomes significant only when the separation is highly likely to be much smaller than the width of the point spread function (PSF). It is important to note that both the direct imaging and superresolution methods use the same prior, ensuring a fair comparison. Despite this, the superresolution method exhibits superior performance. Furthermore, despite different priors, we will implement the same measurement; thus, the measurement itself does not depend on the prior information. It is the performance of measurement (and possibly the downstream analysis) that depends on the prior, whereas the measurement procedure itself does not require any prior knowledge. The prior can be interpreted as a probability distribution of possible separations between the sources. In this view, the overall precision corresponds to a weighted average of the precisions at different separations, with the weights given by the prior probability of each separation. This is particularly relevant in subdiffraction imaging, where the precision can vary significantly depending on the actual separation. When the separation is much smaller than the width of the PSF, direct imaging typically performs poorly and superresolution offers a significant advantage. The superresolution approach can still provide a good estimate of the separation between two point sources even when their distance is not much smaller than the width of the PSF; however, in such cases, its advantage over direct imaging may not be apparent.

It is worth noting that in standard parameter estimation frameworks based on Fisher information, precision is evaluated locally at specific parameter values—that is, the Fisher information quantifies the precision exactly at the true value of the unknown parameter. Of course, in practice, the true value is not exactly known; we typically have only a rough prior distribution, as we assume in our setting. This is a common limitation of Fisher information based precision analysis in the existing discussion of superresolution. What we actually seek is the overall precision when the separation varies across a range of possible values. This is not well captured by a simple Fisher information analysis. In contrast, the quantum learning approach naturally incorporates such prior knowledge: the overall precision is defined as a weighted average of local precisions across the parameter space. Therefore, incorporating the prior distribution should

be seen as a useful advantage of this approach over conventional parameter estimation frameworks.

We also emphasize that the REC formalism quantifies relative error. Therefore, even if the prior indicates that the separation between two point sources is much smaller than the width of the PSF, this information alone is insufficient to guarantee a low error. We believe that relative error is a reasonable choice. In subdiffraction imaging, where the separation is typically very small, relative error offers a more meaningful measure of precision than absolute error. In practice, knowing that the separation is much smaller than the width of the PSF does not imply that we have accessed subdiffraction information or achieved subdiffraction resolution. For example, direct imaging can readily reveal that two point sources are too close to be resolved—simply by observing that their images significantly overlap. This alone provides the prior knowledge that the separation is much smaller than the PSF width. The key point is that recognizing the sources are too close is much easier than actually extracting precise subdiffraction information.

We have added one sentence above Eq. 6 in the main text to briefly mention this: “Assuming $\gamma \ll \sigma$ to exhibit the advantage of superresolution within the Rayleigh limit...”

We also add one paragraph in Sec. II of the Supplemental Material to give full clarification: “We use the prior distribution $p(L) = \frac{1}{\sqrt{2\pi}\gamma} \exp(-\frac{L^2}{2\gamma^2})$, ... we believe this prior is a well-motivated choice for the present discussion. ”

For the more general source considered in our work, we intend to address a broad range of possible prior knowledge. To this end, the prior distribution is randomly generated. For the discussion of both single and multiple generally distributed compact sources, we randomly generate a set of W one-dimensional images. Each image corresponds to a general source of size L , represented by N_{\max} discrete pixels. A random value between 0 and 1 is assigned to each pixel, after which the resulting intensity distribution $I(u)$ is normalized such that $\sum_u I(u) = 1$. By repeating this procedure W times, we obtain a set of W randomly generated one-dimensional intensity profiles. Each image is then used to calculate a corresponding moment vector \vec{x} , which serves as the input parameter in our analysis. The prior probability distribution $p(\vec{x})$ of the moment vectors \vec{x} is defined by the empirical statistics obtained from this set of images. To incorporate this prior knowledge into our calculations, we evaluate the quantities $g_{n_1 n_2} = \int d\vec{x} p(\vec{x}) x_{n_1} x_{n_2}$ and $d_n = \int d\vec{x} p(\vec{x}) x_n$, both of which are essential for the analysis. To obtain the matrix elements d_n and $g_{n_1 n_2}$, we use the set of W moment vectors, denoted as $\vec{x}(w)$, where w indexes each generated sample. We then compute the averages $d_n = \frac{1}{W} \sum_w x_n(w)$ and $g_{n_1 n_2} = \frac{1}{W} \sum_w x_{n_1}(w) x_{n_2}(w)$. This approach provides an effective and flexible way to incorporate general prior information into the estimation process. The figures shown in the main text are based on one such randomly generated prior distribution $p(\vec{x})$. In Section VI of the Supplemental Material, particularly Fig. 17, we verify that different realizations of the random prior do not significantly affect the scaling of the eigenvalues β_k^2 or the behavior of the total REC C_T .

We have revised the corresponding statement regarding the generation of prior knowledge on Page 4 of the main text: “We choose the prior distribution for the moment vectors \vec{x} by randomly generating a set of images and assuming they occur with equal probability, thereby establishing $p(\vec{x})$ as the empirical distribution of the resulting moment vectors, as detailed in Sec. VI of the Supplemental Material.”

We have revised the corresponding statements about the generation of prior knowledge in the two paragraphs of Sec. VI in the Supplemental Material: “To perform the numerical calculations for both single and multiple compact sources,...The intensities are normalized such that the total intensity across all compact sources sums to one. ”

Referee’s comment:

Commenting Fig. 2 (b) the authors say that the fact that the total REC increases by two instead of one at each step is expected. Do they expect this because of the scaling of the eigenvalues in fig 2 (a), or is there

a more intuitive explanation?

Our reply:

Yes, we believe that the total REC C_T increases by two rather than one due to the eigenvalue scaling shown in Fig. 3 of the Supplemental Material (original Fig. 2(a) of the main text), which provides the most straightforward explanation of this effect.

The total REC is given by $C_T = \sum_{k=0}^{K-1} \frac{1}{1+\beta_k^2/S}$, where β_k^2 are the eigenvalues and S is the sample size. When $S \gg \beta_k^2$, the term $\frac{1}{1+\beta_k^2/S} \approx 1$, whereas when $S \ll \beta_k^2$ and $\beta_k^2 \gg 1$, the term $\frac{1}{1+\beta_k^2/S} \ll 1$. Intuitively, the eigenvalues β_k^2 essentially determines the sample size scale at which an incremental increase in C_T occurs, and the magnitude of this increase.

The eigenvalue scaling for direct imaging is shown as $\beta_0^2 = 0$, $\beta_1^2 = \Theta(\alpha^{-2})$, $\beta_2^2 = \Theta(\alpha^{-4})$, and so on. In contrast, the superresolution method exhibits a different pattern: $\beta_0^2 = 0$, $\beta_1^2 = \Theta(\alpha^{-2})$, $\beta_2^2 = \Theta(\alpha^{-2})$, $\beta_3^2 = \Theta(\alpha^{-4})$, $\beta_4^2 = \Theta(\alpha^{-4})$, etc. Because the superresolution method has two eigenvalues at each scaling order (e.g., two at $\Theta(\alpha^{-2})$, two at $\Theta(\alpha^{-4})$), we observe that C_T increases roughly by two at each transition point, compared to an increase of one for direct imaging.

We have added an explanation when referring to Fig. 2 of the main text (original Fig. 2(b) of the main text) for clarity: “The total REC $C_T = \sum_k \frac{1}{1+\beta_k^2/S}$ shows that each β_k^2 sets the sample size at which its eigentask contributes significantly, with contributions near 1 when $S \gg \beta_k^2$ and negligible when $S \ll \beta_k^2$.”

Referee’s comment:

Commenting the results in Fig. 2 the authors say that the saturates at value which is approximatively L/σ . But the plots show a value that is systematically larger. Did the authors tried to estimate this value? Also, the plots are generated for specific values of L fixing σ . Did the authors tried to use different values of σ and L , but keeping the ratio fixed to justify their claim that L/σ is the important quantity here?

Our reply:

The intuition behind the approximation $C_T \approx L/\sigma$ stems from the idea that the total size of the source is L , while the imaging resolution is limited by the width σ of the PSF. Thus, each segment of the source of size σ can be roughly treated as an independent degree of freedom in the imaging system. This suggests that the total number of degrees of freedom—and hence the total REC—should roughly scale as L/σ .

However, this estimate is quite rough and serves only as an intuitive argument rather than a rigorous treatment. It is not strictly accurate to assume that each resolvable segment of the source has size exactly equal to σ , since σ is the width of the Gaussian PSF used in Fig. 5(a) of the Supplemental Material (original Fig. 2(c) in the main text), which lacks a sharp cutoff. There is some overlap between nearby Gaussian envelopes corresponding to different points on the source, which varies depending on whether the point lies near the edge or the center of the source. Therefore, we expect the total REC to scale as $C_T \approx cL/\sigma$, where c is a constant of order $\Theta(1)$. We have clarified this point further in our discussion of in Fig. 5(a) of the Supplemental Material (original Fig. 2(c) in the main text).

The ratio $C_T/(L/\sigma)$ is explicitly examined in Fig. 6 of the Supplemental Material. The figure shows that this ratio remains on the order of $\Theta(1)$ across different values of L and σ , but it exhibits a slight decreasing trend as L increases. In Fig. 7 in the Supplemental Material, we also plot the total REC C_T as a function of σ , where the numerical results indicate that C_T approximately scales as $C_T \propto \sigma^{-0.9}$ for the model considered here.

Since we now provide a new concrete simulation example to explain the physical meaning of the quantum learning formalism in the newly added section “Demonstrative example,” we have moved the original discussion of Fig. 2(c) to Sec. IV of the Supplemental Material.

We have changed the main text where we refer to Fig. 5(a) of the Supplemental Material (original Fig. 2(c) in the main text): “The total REC represents the number of degrees of freedom we can extract from the measurement, and since the image is blurred, it is roughly on the order of L/σ ; however, this relation is not exact, as it arises from heuristic reasoning rather than formal analysis.”

We also updated Fig. 6 and 7 in the Supplemental Material and the corresponding text in Sec. IV of the Supplemental Material: “We also demonstrate that the total REC C_T increases approximately linearly with the source size L , ... it provides a rough estimate that serves only as an intuitive argument rather than a rigorous treatment.”

Referee’s comment:

The caption of Fig. 5 seems wrong. It suggests that the centroids of the two extended sources are fixed, but the size is changed. But from the plot (and the main text) it seems that the size of the sources is fixed, and their distance is changed. The authors should correct or clarify what they are showing.

Our reply:

Figure 5 of the main text illustrates the case where the source size is fixed and the centroid separation is varied.

We have updated the caption of Fig. 5 to clarify this point: “Here, $q = 1, 2$ corresponds to the case of two compact sources ($Q = 2$) with centroids located at $\pm L/4$, so that the separation between the two sources is $L/2$.”

We have also updated the main text in Page 7 to clarify this point: “...considering different distances between the centroids of the two compact sources. Note that the construction of the basis states defined in Eq. 12 does not depend on the size of each source L_q .”

Referee’s comment:

Commenting Fig. 5 in the text, the authors refer to Hermite Gaussian mode sorters and say that in Sec V D of the supplementary they show that these sorters can be used to implement orthogonalized SPADE. I couldn’t find this information there. In fact, all the references about the mode sorters are included in the bibliography of the supplementary material but are not cited in the text. Either the statement in the main text should be removed, or the supplementary material should be amended.

Our reply:

The key idea behind using the Hermite-Gaussian (HG) mode sorter to implement the orthogonalized SPADE method is that the mode sorter enables a transformation from HG modes $|b_{q,m}\rangle$ to transverse momentum eigenstates $|m\rangle$, which can be expressed as $|m\rangle = F_q|b_{q,m}\rangle$, where F_q denotes the transformation performed by the mode sorter. Each HG mode is defined as $|b_{q,m}\rangle = \int du b_{q,m}(u)|u\rangle$, where $b_{q,m}(u)$ is the m th-order HG mode centered at the centroid of the q th source.

In our implementation of the orthogonalized SPADE method, the measurement projects onto the POVM elements $|\phi_{j\pm}^{(l)}\rangle = \frac{1}{\sqrt{2}} \left(|b_j^{(l)}\rangle \pm |b_j^{(l+1)}\rangle \right)$, with $|b_j^{(l)}\rangle = \int du b_j^{(l)}(u)|u\rangle$. Crucially, the basis functions $b_j^{(l)}(u)$ used in this method can be expressed as linear combinations of the HG modes $b_{q,m}(u)$. Therefore, the

mode sorter F_q enables the transformation of the orthogonalized basis $b_j^{(l)}(u)$ in Eq. (13) of the main text into simple momentum states (up to additional post-processing), thereby allowing full implementation of the desired measurement.

For example, consider the imaging of two compact sources using the orthogonalized SPADE method. One of the basis vectors is given by

$$\begin{aligned}
 b_1^{(1)}(u) &= p_3 b_{1,0}(u) + p_4 b_{2,0}(u) + p_5 b_{1,1}(u), & p_3 &= \frac{\sqrt{4\sigma^2 + \frac{\frac{u_1 u_2}{2\sigma^2} (u_1 - u_2)^2}{e \frac{u_1 u_2}{2\sigma^2} - e \frac{u_1^2 + u_2^2}{4\sigma^2}} (u_1 - u_2)}}{4 \left(-1 + e \frac{(u_1 - u_2)^2}{4\sigma^2} \right) \sigma^2 - (u_1 - u_2)^2}, \\
 p_4 &= \frac{e \frac{(u_1 - u_2)^2}{8\sigma^2} \sqrt{4\sigma^2 + \frac{\frac{u_1 u_2}{2\sigma^2} (u_1 - u_2)^2}{e \frac{u_1 u_2}{2\sigma^2} - e \frac{u_1^2 + u_2^2}{4\sigma^2}} (-u_1 + u_2)}}{4 \left(-1 + e \frac{(u_1 - u_2)^2}{4\sigma^2} \right) \sigma^2 - (u_1 - u_2)^2}, & p_5 &= -\frac{2 \left(-1 + e \frac{(u_1 - u_2)^2}{4\sigma^2} \right) \sigma \sqrt{4\sigma^2 + \frac{\frac{u_1 u_2}{2\sigma^2} (u_1 - u_2)^2}{e \frac{u_1 u_2}{2\sigma^2} - e \frac{u_1^2 + u_2^2}{4\sigma^2}}}}{4 \left(-1 + e \frac{(u_1 - u_2)^2}{4\sigma^2} \right) \sigma^2 - (u_1 - u_2)^2}.
 \end{aligned} \tag{3}$$

where $u_{1,2}$ denote the centroids of the two compact sources, σ is the width of a Gaussian PSF. As long as the Hermite-Gaussian modes $b_{q,m}$ can be converted into momentum states, only an additional beam splitter is needed to implement the projection onto the POVM with postselection. This procedure is explicitly illustrated in Fig. 11 of the Supplemental Material.

We have revised the description in the Sec. V D of the Supplemental Material to clarify the implementation of the method. Specifically, we added the statement: “In this section, we discuss how to implement the orthogonalized SPADE method using a Hermite-Gaussian mode sorter, beam splitters, and postselection... We illustrate this implementation with a representative example: imaging two compact sources.” and later explained: “These basis functions, constructed under the assumption of a Gaussian point spread function, are the Hermite-Gaussian modes... the basis functions $b_j^{(l)}(u)$ can be computed as linear combinations of the Hermite-Gaussian modes”

We have included all the citations from the Supplemental Material on Page 7 of the main text: “In Sec. V of the Supplemental Material, we demonstrate that the Hermite-Gaussian mode sorter [38-42] can be used to implement the orthogonalized SPADE method with some additional steps for a Gaussian PSF, and we also provide more details on the derivation of the scaling of β_k^2 and the eigenvectors r_k based on numerical calculations.”

Report of the First Referee

Referee's comment:

The authors have addressed my previous concerns and improved the clarity of the manuscript. I can recommend its publication.

Our reply:

We thank the referee for recommending our work for publication in Nature Communications and for the valuable feedback that helped us improve the manuscript.

Report of the Second Referee

Referee's comment:

I have now reviewed both the revised submission and detailed responses of the authors to all reviewers' comments, suggested corrections, and recommendations for the manuscript. The manuscript is now substantially improved in its clarity and rigor of presentation and flow, with the various statements and arguments largely well supported by additional mathematical calculations, figures, improved figure captions, and qualitative statements included either in the manuscript or its supplementary materials. It is unfortunate that the supplementary materials are far more voluminous and detailed than the main paper, which still lends a certain opaqueness to the arguments in the main text, but I do believe this is about the best balance that could be struck without making the main paper too bulky and difficult to follow. The authors have painstakingly addressed all my original review comments in a satisfactory manner, and I therefore recommend the publication of the manuscript in Nature Communications in its revised form.

Our reply:

We thank the referee for recommending our work for publication in Nature Communications and for the valuable feedback that helped us improve the manuscript.

Report of the Third Referee

Referee's comment:

The manuscript was significantly rewritten and now it is more accessible to readers without a strong background in learning theory, and it is far more readable without continuously looking into the supplementary material.

My main criticism was that the previous version lacked a practical interpretation of the REC results. The authors addressed this point by improving their presentation and adding a "Demonstrative Example" section. This is certainly helpful, however I find that there is a dissonance between the example they choose and the criticism they raise to the QFIM approach.

I agree with the authors that when performing imaging tasks, i.e. reconstructing the shape of the objects, if we want to use estimation theory (i.e. the QFIM matrix approach) we need to choose a parametrization of the object, and that finding the best one is far from trivial. I also agree with the authors that in this context a learning approach that finds by itself the right parametrization is very attractive especially for complex scenes or in experimental situations where it is impossible to fully characterize the system.

On the other hand, the classification task with QR-code-like objects studied by the authors is not an image reconstruction task, but an hypothesis testing task, and as such it would be unnatural to analyze it with estimation theory. There is quite some literature on using SPADE for hypothesis testing tasks, in particular Grace, and Guha, PRL 129, 180502 (2022) proved the optimality of SPADE for arbitrary multiple hypothesis testing tasks, and identified bounds on the error probabilities for both symmetric and asymmetric hypothesis testing scenarios. They also considered the sorting of QR-codes. For such a task likelihood ratio kind of tests are known to achieve optimal results. It would be extremely interesting if the authors could compare the error probabilities achieved with their approach with those ultimate bounds. Can the authors reach the ultimate bounds with their approach? Does their approach need a less precise characterization of the system to perform the testing since the system can learn by itself?

If the authors could answer positively to these questions, and clarify that with little increased complexity their approach could generalize to more complex tasks like face recognition, or image reconstruction, then this work would definitely deserve to be published in nature communication. If not, I think it would not meet the impact criteria of this journal, and would be more appropriate for a more technical one.

Our reply:

We thank the referee for the valuable comments. We have answered all of the questions positively, carried out all of the calculations requested by the referee, and revised the manuscript accordingly. Our detailed responses are provided below, with the corresponding changes highlighted in blue in both the main text and the Supplemental Material and described below.

Face recognition example

We have added a new face-recognition example to illustrate both our learning-based approach and the advantages of the orthogonalized SPADE method. This example appears in a newly added Sec. VIII C of the Supplemental Material, and we have also updated the section "Demonstrative example" in the main text to incorporate this stronger example. To the best of our knowledge, this is the first face-recognition example in

the context of superresolution based on a SPADE-type method. It highlights the versatility and effectiveness of the learning-based approach in handling complex imaging problems and represents an important step from fundamental analysis toward making superresolution techniques more practically applicable.

The face images used in our simulation are taken from the Olivetti Faces dataset provided by AT&T Laboratories Cambridge. Each individual has 64×64 grayscale images with different facial expressions and small variations in pose. In our simulation, we construct a training set and a testing set using 20 individuals. To enable face recognition, we follow exactly the same machinery as in the previous examples. We first construct the eigentasks, which identify the well-estimated features together with their sample thresholds determined by the corresponding eigenvalues. These well-estimated eigentasks are then used as inputs to train a multi-class classifier based on a multinomial logistic (softmax) model. The trained classifier is subsequently used to perform inference on the test set. This example exhibits the same behavior observed earlier: the total REC C_T provides a reliable guideline for determining how many eigentasks should be retained for both training and inference. Moreover, the orthogonalized SPADE method shows clear advantages in success probability compared to the other two methods.

This face-recognition example clearly demonstrates the advantage of the learning-based approach in the context of superresolution. The method identifies well-estimated features directly from the structure of the problem and the underlying quantum measurement, thereby capturing all information extractable from the measurement and the state. Extending the previous examples to this face-recognition task requires only changing the input sources and replacing the binary logistic regression model with its multiclass version. All of the complex structure in the face images is then automatically handled through the eigentask-identification and training procedure. Thus, as the referee noted, our learning-based approach requires a much less precise characterization of the system, because the system can identify the relevant features on its own. In many practical scenarios, this is crucial: obtaining an accurate characterization of the system, such as in the face-recognition example considered here, is extremely challenging.

Comparison with Chernoff bound and likelihood-ratio method

We have added a new Sec. IX in the Supplemental Material that presents detailed simulations and comparisons among the learning-based approach, the likelihood-ratio method, and the Chernoff bound. A brief summary of this discussion has also been incorporated into the main text: “Many machine-learning tasks, including face recognition, ... Simulated examples illustrating the above discussion are provided in Sec. IX of the Supplemental Material.”

Some machine-learning tasks can indeed be formulated as discrimination problems, but this does not trivialize the usefulness of learning-based approaches. Even face recognition can be viewed as a discrimination problem whose ultimate performance is bounded by the Chernoff bound and, in principle, solvable by the likelihood-ratio method if a complete statistical model were available. In practice, however, face recognition is a prototypical example where such a model is far too complex, and where learning-based methods offer a clear practical advantage—reflected in their widespread use in real-world systems.

When compared with the likelihood-ratio method, a key advantage of learning-based approaches is their ability to handle highly complex image structures. In contrast, the likelihood-ratio method typically requires an accurate statistical model. For example, in a face-recognition task involving tens of individuals, it is extremely challenging to write down a closed-form likelihood function for the measured data that would enable an exact likelihood-ratio calculation. Quantum-learning-based approaches naturally excel in such complex scenarios, as they exploit the structure of quantum measurements and quantum states through the learning procedure, enabling them to operate effectively in the complex imaging tasks considered here.

Having clarified that learning-based approaches offer clear benefits for complex tasks such as face recognition—where an exact likelihood-ratio calculation is not feasible—we now turn to a simpler setting.

In the newly added simulation of distinguishing a single point source from two point sources in Sec. IX of the Supplemental Material, we find that the likelihood-ratio method and the learning-based method perform almost identically across all tested regimes. Moreover, when the number of samples becomes sufficiently large, both approaches converge toward the Chernoff bound.

When compared with the Chernoff bound, it is true that a learning-based method cannot surpass this limit, as the Chernoff bound represents the fundamental asymptotic bound for discrimination. However, we emphasize that the Chernoff bound is far from a complete characterization of performance in many discrimination problems. Its limitations already manifest in the simple example of distinguishing a single point source from two point sources, as shown in the newly added simulation in Sec. IX of the Supplemental Material. In this example, the exponential index $\zeta = -\log(1 - P_{\text{succ}})$, where P_{succ} is the success probability, only begins to follow the scaling predicted by the Chernoff bound once P_{succ} has already exceeded 99%. In other words, the Chernoff bound becomes accurate only in the regime where the task is almost guaranteed to succeed. In practical settings, however, one is typically concerned not with how rapidly P_{succ} approaches unity once it is already above 99%. Instead, the relevant question is when a moderate success probability—say, 70%—is reached and what strategy should be adopted to achieve it. Such nonasymptotic behavior is clearly not captured by the Chernoff bound in our simulation. And more importantly, the Chernoff bound provides no indication of which strategy should be used when the likelihood-ratio method is not tractable, as is the case in many complex imaging problems. By contrast, our learning-based approach provides the total REC as a meaningful figure of merit in the finite-sample regime and, more importantly, offers a principled and practical strategy for discrimination tasks when the number of collected samples S is finite.

For the complex tasks such as the newly added face-recognition example in the main text and Sec. VIII C of the Supplemental Material, a direct comparison with the likelihood-ratio method becomes impractical: we do not know how to construct the likelihood function, and doing so is generally infeasible for realistic imaging problems. Moreover, because we are not operating in the regime where the success probability approaches 100%, the Chernoff bound does not provide a meaningful benchmark. Nevertheless, we include a simulation of the success probability for this face-recognition task in Sec. IX of the Supplemental Material, which exhibits a step-like dependence on α at fixed sample number. Importantly, the success probability in this regime has not yet entered the vicinity of 100%, and the learning-based approach allows us to probe performance well before the asymptotic regime characterized by the Chernoff bound while also offering a principled strategy for tackling complex discrimination problems with finite samples.

Also note the Chernoff-bound framework discussed in [Grace and Guha, PRL 129, 180502 (2022)] does not appear capable of handling the face-recognition scenario considered here. Their analysis generalizes earlier superresolution work—extending the discrimination of simple point-source configurations, such as in [Huang and Lupo, PRL 127, 130502 (2021)], to distinguishing among M different general objects. This generalization already represents a substantial and technically involved theoretical development, leading to a full PRL publication based on this analysis alone. In our setting, however, computing the Chernoff bound would require addressing a significantly more difficult problem. Instead of discriminating among M fixed objects, one would need to discriminate among M sets of images, where each set corresponds to an individual and effectively contains an infinite number of possible face images for that person. Given any new image, the task is to assign it to one of these M sets. Computing the corresponding Chernoff exponents for these image sets would be considerably more complex than the case considered in [Grace and Guha, PRL 129, 180502 (2022)] and lies far beyond the scope of the present work. This complexity deserves a dedicated theoretical treatment in its own right. Moreover, since the Chernoff bound is not a useful figure of merit in the finite-sample regime considered in our work, as discussed above, this should not affect the delivery or interpretation of our main results.

We thus confirm the referee’s point: (1) The learning-based method can indeed approach the Chernoff

bound, provided that the sample number is sufficiently large, just as the likelihood-ratio method does. However, this convergence becomes less meaningful in some cases, because the Chernoff bound is only reached when the success probability is already extremely close to 100% for both the likelihood-ratio method and the learning-based method. (2) The learning-based method requires a far less precise characterization of the system, since it can infer the relevant structure directly from the data. In fact, for complex practical imaging tasks, this becomes a major advantage: the learning approach can identify the structure of the image without needing an explicit statistical model.

We have also revised several parts of the introduction to incorporate these discussions of the likelihood-ratio and Chernoff-bound approaches, as well as the newly added face-recognition examples. Rather than framing the discussion around the limitations of the Fisher-information approach alone, we now present the broader challenges faced by conventional statistical tools. (i) Modeling complexity and ambiguity. In general, the natural parameterization of a complex source is unknown. The eigentasks identified by our quantum-learning approach correspond precisely to the well-estimated features, resolving this ambiguity. In addition, the likelihood-ratio method requires a complete statistical model of the object, which can be extremely challenging to obtain in practice. In the learning-based approach, using the features identified by eigentasks together with the training procedure removes the need to construct a closed-form likelihood function and enables complex practical tasks to be handled effectively. (ii) Finite-sample reliability. Effective use of the data requires isolating and retaining only the well-estimated features—a particularly difficult task in imaging, where the number of parameters is, in principle, infinite. In the learning-based approach, this is accomplished by identifying the sample threshold for estimating each feature revealed by the eigentask. Neither Fisher-information nor Chernoff-bound analyses provides such a concrete strategy in the finite-sample regime relevant to practical imaging tasks.

The revised manuscript now includes face recognition as a substantially more complex and practically relevant example demonstrating the impact of our learning-based approach. We have also incorporated detailed discussions and simulations highlighting the advantages of learning-based approach over conventional statistical tools, including the Chernoff bound and the likelihood-ratio method. We hope the referee finds the revised manuscript significantly improved and suitable for publication in Nature Communications.

Report of the Third Referee

Referee's comment:

The authors addressed all my concerns. They proved that for simple tasks the learning approach is equivalent to optimal statistical methods. Moreover, the learning approach can be applied to complex tasks where a full statistical model of the problem is difficult if not impossible to formulate, and provide satisfactory results. The paper in this form can be published in nature communication.

Our reply:

We thank the referee for recommending our work for publication in Nature Communications and for the valuable feedback that helped us improve the manuscript.